# Revealing the Illusion of Joint Multimodal Understanding in VideoQA Models

## Abstract

While VideoQA Transformer models demonstrate competitive performance on standard benchmarks, the reasons behind their success are not fully understood. Do these models jointly capture and leverage the rich multimodal structures and dynamics from video and text? Or are they merely exploiting shortcuts to achieve high scores? Hence, we design *QUAG* (QUadrant AveraGe), a lightweight and non-parametric probe, to critically analyze multimodal representations. QUAG facilitates combined dataset-model study by systematic ablation of model's coupled multimodal understanding during inference. Surprisingly, it demonstrates that the models manage to maintain high performance even under multimodal impairment. We extend QUAG to design "QUAG-attention", a simplistic and less-expressive replacement of self-attention. We find that the models with QUAG-attention achieve similar performance with significantly less mulops without any finetuning. These findings indicate that the current VideoQA benchmarks and metrics do not penalize models that find shortcuts and discount joint multimodal understanding. Motivated by this, we propose *CLAVI* (Counterfactual in LAnguage and VIdeo), a diagnostic dataset for coupled multimodal understanding in VideoQA. CLAVI consists of temporal questions and videos that are augmented to curate balanced counterfactuals in language and video domains. We evaluate models on CLAVI and find that all models achieve high performance on multimodal shortcut instances, but most of them have very poor performance on the counterfactual instances that necessitate joint multimodal understanding. Overall, with the multimodal representation analysis using QUAG and diagnostic analysis using CLAVI, we show that many VideoQA models are incapable of learning multimodal representations and that their success on standard datasets is an illusion of joint multimodal understanding.

## 1 Introduction

Multimodal learning with videos and language is challenging, despite the shared sequential nature of these modalities, due to their distinct underlying structures. That is, videos exhibit spatio-temporal dynamics in the pixel space, whereas language representation is composed of the syntax and semantics of word sequences. Hence, tasks like Video Question Answering (VideoQA) (Zhong et al., 2022) are difficult as they necessitate the model to acquire accurate representations of both the modalities and establish meaningful connections between them. Transformers have demonstrated exceptional performance on VideoQA benchmarks (Zhong et al., 2022). However, since they lack the intrinsic inductive biases for these representation, they must learn it from the data (Xu et al., 2021; Patrick et al., 2021). But does the good performance of Transformers on current VideoQA benchmarks necessarily mean that they learn to faithfully represent, leverage, understand and reason the modalities? Or do the current benchmarks and metrics fail to robustly evaluate the models for their multimodal understanding?

This is a valid concern because deep learning models can learn shortcuts to achieve good performance without faithfully representing underlying modalities (Geirhos et al., 2020). For example, seemingly spatio-temporal tasks, like some action classification problems, are shown to be solved without focusing much on temporal representations (Kowal et al., 2022; Sevilla-Lara et al., 2021). Similarly, in VideoQA, recent works report that the datasets contain specific biases (Buch et al.,

2022; Lei et al., 2023). However, these works are restricted to isolated analyses of either the models or the datasets. This raises questions: **Are the models actually learning to jointly leverage and understand the modalities, or is the performance on the current benchmarks an illusion of joint multimodal learning?**

To answer these questions, we propose QUadrant AveraGe (QUAG), a lightweight and non-parametric probe to systematically gauge the reliance of a finetuned model's performance on joint multimodal representations. We posit that joint multimodal understanding is enabled in the fusion layers by progressively attending to the informative tokens within and between the modalities. QUAG impairs the components of modality fusion by block-averaging attention weights. We apply QUAG on multiple dataset-model combinations, and consistently find that the models manage to achieve high performance on the benchmarks without relying specific multimodal interactions.

This finding is concerning because high performance on established benchmarks should be ideally indicative of coupled multimodal understanding. We establish that these models learn sub-optimal representations; that is, the modality fusion doesn't effectively capture the information within each modality along with the complementary information in the other modality. We validate the sub-optimality in multimodal representations by replacing self-attention in the pretrained models with simple and less-expressive QUAG-attention. Even though QUAG-attention impairs multimodal capabilities, the models augmented with QUAG-attention manage to maintain the high performance on standard benchmarks without any finetuning. This raises a follow-up question – **How then can we diagnose coupled multimodal understanding?**

Thus, we create Counterfactual in LAnguage and VIsion (CLAVI), a diagnostic benchmark to robustly assess joint multimodal understanding in VideoQA models. Temporal understanding ideally requires coupled multimodal understanding. However, the standard benchmarks do not contain or assess performance on counterfactual instances. CLAVI contains automatically generated balanced temporal counterfactuals in both question and video domains to accurately test if the models can jointly understand temporal cues in the question (temporal prepositions and adverbs) and the video (order of frames) domains (Figure 2). We develop consistent-accuracy metrics to precisely assess the contributions of shortcuts to circumvent joint multimodal understanding. We find that finetuned models have high-accuracy on shortcut instances in CLAVI, but have poor performance on the counterfactual instances that require coupled multimodal understanding. Hence, we position CLAVI as a litmus test to diagnose joint multimodal understanding which is overlooked by the existing datasets.

In summary, our contributions are *(i)* we develop QUAG, a systematic method to identify sub-optimalities in joint multimodal representations, *(ii)* using QUAG and QUAG-attention, we demonstrate that high performance on established VideoQA benchmarks is not representative of faithful coupled multimodal understanding, and *(iii)* we develop CLAVI, a new diagnostic benchmark that contains balanced temporal counterfactuals in videos and questions to confidently disambiguate the contributions of shortcuts in joint multimodal learning to benchmark the models. Overall, QUAG and CLAVI provide holistic dataset-model insights that reveal the illusion of multimodal understanding in VideoQA models.

## 2 DO VIDEOQA MODELS LEARN TO JOINTLY LEVERAGE THE MODALITIES?

We posit that coupled multimodal understanding is enabled in the fusion layers by progressively attending to the informative tokens within and between the modalities. Hence, we design QUAG to systematically ablate the effects of multimodal attention. It impairs the joint multimodal representations in the pretrained model by systematically block-averaging the attention weights to attend to all tokens uniformly at inference time. Based on the targeted modality-interactions, we define special cases of QUAG, collectively called short-circuit operations and analyze the performance drop.

### 2.1 VIDEO QUESTION ANSWERING SETUP

In VideoQA, the task is to predict the correct answer given a video-question tuple, $(\mathcal{V}, \mathcal{T})$. A VideoQA model consists of a vision encoder $\boldsymbol{F}_{\mathcal{V}} : \mathcal{V} \to \mathbb{R}^{l_{\mathcal{V}} \times d}$, text encoder $\boldsymbol{F}_{\mathcal{T}} : \mathcal{T} \to \mathbb{R}^{l_{\mathcal{T}} \times d}$, and a multimodal fusion module $M : (\boldsymbol{F}_{\mathcal{V}}(\mathcal{V}), \boldsymbol{F}_{\mathcal{T}}(\mathcal{T})) \to \mathbb{R}^{(l_{\mathcal{V}} + l_{\mathcal{T}}) \times d}$, where $l_{\mathcal{V}}$ and $l_{\mathcal{T}}$ are the maximum input sequence lengths of video and text modalities respectively and $d$ is the dimensionality of the fusion model.

Consider $M$ as a composition of $n$ attention-based multimodal fusion blocks, $M = M_n \circ M_{n-1} \circ \cdots M_1$. Each fusion block consists of attention, normalization, and token-mixing modules. For our analysis, we consider $M$ to be composed of self-attention transformer blocks. That is, query, key, and value are the transformations of the same input sequence. $\boldsymbol{X}_{\mathcal{VT}} = [\boldsymbol{F}_{\mathcal{V}}(\mathcal{V}) \parallel \boldsymbol{F}_{\mathcal{T}}(\mathcal{T})] \in \mathbb{R}^{(l_{\mathcal{V}}+l_{\mathcal{T}}) \times d}$ is the input for $M$, where $\parallel$ is concatenation operator. Since QUAG operates at inference time, we assume the VideoQA model to be finetuned and frozen.

## 2.2 QUAG: Ablation of modality interactions

Shortcuts are the spurious features learned by a given model on a given dataset (Murali et al., 2023). Along this axis, we use QUAG to pinpoint the exact failure modes in the dataset representations learned by the models.

Let $\boldsymbol{X}_{i-1}$ denote the input of the fusion block $M_i$ and let $(\boldsymbol{Q}_i, \boldsymbol{K}_i, \boldsymbol{V}_i)$ be its query, key, and value transformations and $\boldsymbol{X}_0 = \boldsymbol{X}_{\mathcal{VT}}$. Then, the token-mixing operation is given by $\boldsymbol{T}_i = \boldsymbol{A}_i \boldsymbol{V}_i$, where $\boldsymbol{A}_i = softmax(\boldsymbol{Q}_i \boldsymbol{K}_i^\top)$ is the attention matrix (we omit the scaling factor $\sqrt{d}$ for readability). For $\boldsymbol{Q}_{1u}$, $\boldsymbol{K}_{1u}$, and $\boldsymbol{V}_{1u}$ to denote the query, key, and value projections of modality $u$ for the first fusion block, $M_1$, we can simplify, $\boldsymbol{A}_1$ and $\boldsymbol{T}_1$ in terms of their partition blocks, referred to as quadrants henceforth, as:

$$\boldsymbol{A}_1 = softmax\left(\left[\begin{array}{c|c} \boldsymbol{Q}_{1\mathcal{V}}\ \boldsymbol{K}_{1\mathcal{V}}^\top & \boldsymbol{Q}_{1\mathcal{V}}\ \boldsymbol{K}_{1\mathcal{T}}^\top \\ \hline \boldsymbol{Q}_{1\mathcal{T}}\ \boldsymbol{K}_{1\mathcal{V}}^\top & \boldsymbol{Q}_{1\mathcal{T}}\ \boldsymbol{K}_{1\mathcal{T}}^\top \end{array}\right]\right) \quad \text{and} \quad \boldsymbol{T}_1 = \left[\begin{array}{c|c} \boldsymbol{A}_{\mathcal{VV}}^1 & \boldsymbol{A}_{\mathcal{VT}}^1 \\ \hline \boldsymbol{A}_{\mathcal{TV}}^1 & \boldsymbol{A}_{\mathcal{TT}}^1 \end{array}\right]\left[\begin{array}{c} \boldsymbol{V}_{1\mathcal{V}} \\ \hline \boldsymbol{V}_{1\mathcal{T}} \end{array}\right]$$

where $\boldsymbol{A}_{u_1 u_2}^1$ represents the quadrant of $\boldsymbol{A}_1$ corresponding to $(\boldsymbol{Q}_{1u_1} \boldsymbol{K}_{1u_2}^\top)$. Note that we skip layer normalization layers in the discussion for simplicity. Hence, we can simplify and write $\boldsymbol{T}_1$ as:

$$\boldsymbol{T}_1 = \left[\begin{array}{c} \boldsymbol{A}_{\mathcal{VV}}^1 \boldsymbol{V}_{1\mathcal{V}} + \boldsymbol{A}_{\mathcal{VT}}^1 \boldsymbol{V}_{1\mathcal{T}} \\ \hline \boldsymbol{A}_{\mathcal{TV}}^1 \boldsymbol{V}_{1\mathcal{V}} + \boldsymbol{A}_{\mathcal{TT}}^1 \boldsymbol{V}_{1\mathcal{T}} \end{array}\right] \tag{1}$$

We follow the same partition quadrants, as defined for $\boldsymbol{A}_1$ in $M_1$, for $\boldsymbol{A}_j$ in the downstream fusion layer $M_j$ and denote the quadrants as $\boldsymbol{A}_{u_1 u_2}^j$. Next, we define row-wise average and replace operator $\mathcal{R}$ that operates on a quadrant of a matrix to replace the values in the quadrant with the mean value of the respective partitioned-row. Note that the values in the other quadrants are unaffected. Given a matrix $\boldsymbol{Z}$ of size $p \times q$ and let $W$ denote the location of the quadrant of $\boldsymbol{Z}$ with indices $(p_1^W \cdots p_2^W) \times (q_1^W \cdots q_2^W)$. We use $[\,.\,]_{ij}$ to index the element in row $i$ and column $j$. Then,

$$[\mathcal{R}(\boldsymbol{Z}, W)]_{ij} = \begin{cases} \sum_{k=q_1^W}^{q_2^W} \frac{[\boldsymbol{Z}]_{ik}}{q_2^W - q_1^W + 1} & i \in \{p_1^W, \cdots, p_2^W\} \text{ and } j \in \{q_1^W, \cdots, q_2^W\} \\ [\boldsymbol{Z}]_{ij} & \text{otherwise} \end{cases}$$

We can now formally define the QUAG operator, $\phi$, as:

$$\phi(\boldsymbol{A}_i, \boldsymbol{V}_i, [s_1, s_2, \cdots, s_m]) = (\mathcal{R}_{s_1} \circ \mathcal{R}_{s_2} \cdots \circ \mathcal{R}_{s_m}(\boldsymbol{A}_i))\boldsymbol{V}_i$$

where $S = [s_1, s_2, \cdots, s_m]$ is a list of quadrants such that $\forall s \in S : s \in \{\mathcal{TT}, \mathcal{TV}, \mathcal{VT}, \mathcal{VV}\}$, $\mathcal{R}_{s_i}(\boldsymbol{Z})$ is short-hand for $\mathcal{R}(\boldsymbol{Z}, s_i)$, $\boldsymbol{A}_i$ and $\boldsymbol{V}_i$ are the attention and value matrices of $M_i$ respectively. Note that $\mathcal{TT}$ refers to the quadrant corresponding to $A_{\mathcal{TT}}^i$ (independent of the index $1 \le i \le n$ of $A$), similarly $\mathcal{TV}$ refers to the quadrant corresponding to $A_{\mathcal{TV}}^i$, and so on. In implementation, we re-adjust the quadrant boundaries to ignore the padded elements. Refer Figure 1 for an illustrative example. Incorporating QUAG in the existing model pipeline is very easy and we provide the code in the Appendix A.2.2. Since we will be applying the QUAG operator successively on all the layers of $M$, for brevity, we denote $\Phi(M, S) = \forall_{1 \le i \le n} \phi(\boldsymbol{A}_i, \boldsymbol{V}_i, S)$. Note that $\phi$, and hence, $\Phi$ is independent of the order of elements in $S$.

## 2.3 Short-circuit operations

As QUAG is a generic method of probing multimodal fusion, we consider some special cases based on the value of $S$ below. We call these operations collectively as short-circuiting operations:

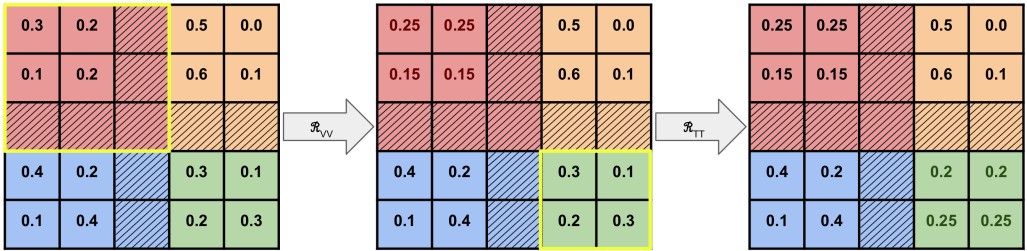

Figure 1: Illustrative toy example of of unimodal short-circuiting or $\phi(\boldsymbol{Z}, [\mathcal{TT}, \mathcal{VV}])$, where $\boldsymbol{Z}$ is the input attention matrix (left-most in the figure), $\mathcal{R}$ is the row-wise average and replace operator and hatching denotes padding. The quadrants that are operated on are highlighted in bright yellow box. Note that $l_{\mathcal{V}} = 3$ and $l_{\mathcal{T}} = 2$ for the model and the video embeddings are pre-concatenated with the question embeddings. As shown in the figure, we apply $\mathcal{R}$ successively to replace the values in the quadrant with the respective row-wise average value. The cells are colored as per their quadrants ($\mathcal{VV}$ : red, $\mathcal{VT}$ : yellow, $\mathcal{TV}$ : blue, $\mathcal{TT}$ : green).

1) $\boldsymbol{S} = [\mathcal{VV}, \mathcal{TT}]$: $\phi(\boldsymbol{A}_1, \boldsymbol{V}_1, [\mathcal{VV}, \mathcal{TT}])$ is equivalent to scaling the average values of $\boldsymbol{V}_{1\mathcal{V}}$ and $\boldsymbol{V}_{1\mathcal{T}}$ in the upper and lower blocks of $\boldsymbol{T}_1$ respectively (as evident from Eqn. 1). Hence, in the upper block, video queries faithfully attend over text keys but uniformly over video keys. Likewise, text queries attend faithfully over video queries but uniformly over text queries in the lower block. We illustrate this operation in Figure 1 and call such a fusion block to be unimodal average conformable.

Having understood the trivial case, we prove by induction that $\Phi(M, [\mathcal{VV}, \mathcal{TT}])$ leads to unimodal average conformability of all the component fusion blocks in $M$. Consider a block $M_j \in M$ such that $j > 1$. We want to show that unimodal average conformability of first $\{M_0, M_1, \cdots, M_{j-1}\}$ blocks using $\forall_{1 \le i \le j-1} \; \phi(\boldsymbol{A}_i, \boldsymbol{V}_i, [\mathcal{VV}, \mathcal{TT}])$ implies $\phi(\boldsymbol{A}_j, \boldsymbol{V}_j, [\mathcal{VV}, \mathcal{TT}])$ will make $M_j$ unimodal average conformable. The input of $M_j$ can be decomposed into non-linear and linear (from the residual connection that skips the feed-forward layer of $M_{j-1}$) projections of $\boldsymbol{T}_{j-1} + M_{j-2} \circ M_{j-3} \cdots \circ M_1(\boldsymbol{X}_{\mathcal{VT}}) + \boldsymbol{X}_{\mathcal{VT}}$. Hence, when $\{M_0, M_1, \cdots, M_{j-1}\}$ are unimodal average conformable, $\boldsymbol{X}_{\mathcal{VT}}$ is the only non-conformable component. And we have shown in the trivial case that $\phi(\boldsymbol{A}_1, \boldsymbol{V}_1, [\mathcal{VV}, \mathcal{TT}])$ makes $M_1$ conformable, hence $M_j$ is also unimodal average conformable under $\phi$.

Ultimately, $\Phi(M, [\mathcal{VV}, \mathcal{TT}])$ bypasses the effect of video-video attention and text-text attention. We prove that unimodal token-mixing is reduced to scaling the average of the modalities. We term this as **unimodal short-circuiting**. It ablates unimodal representations to analyze their dependence on the performance of the models. Since the following cases can be proved similarly using induction, we skip the proofs for conciseness.

2) $\boldsymbol{S} = [\mathcal{VT}, \mathcal{TV}]$: Parallel to unimodal short-circuiting, $\phi(\boldsymbol{A}_1, \boldsymbol{V}_1, [\mathcal{VT}, \mathcal{TV}])$ is equivalent to scaling the average values of $\boldsymbol{V}_{1\mathcal{T}}$ and $\boldsymbol{V}_{1\mathcal{V}}$ in the upper and lower blocks of $\boldsymbol{T}_1$ respectively. Video and text queries faithfully attend to video and text keys respectively while crossmodal attention in video-text is reduced to uniform attention. We term this effect as **crossmodal short-circuiting**. It is complementary to unimodal short-circuiting and assesses the importance of inter-modality token-mixing. It probes if the models actually learns by fusing the information between the two modalities or is it largely driven by unimodal biases within the modalities.

3) $\boldsymbol{S} = [\mathcal{VV}, \mathcal{TV}]$: This is equivalent to removing the effect of individual of video keys, resulting in averaging the components of video modality in the upper and lower blocks of all $\boldsymbol{T}_i$. We call this **video short-circuiting**. Similarly, $\boldsymbol{S} = [\mathcal{TT}, \mathcal{VT}]$ leads to **text short-circuiting**.

## 2.4 QUAG-ATTENTION

Along with an assessment of multimodal understanding, QUAG enables a detailed analysis of token mixing for identifying the sub-optimality of learned representations. Sub-optimality occurs if the fusion process doesn't effectively capture the information within each modality along with the complementary information in the other modality. Hence, we use QUAG as an inspiration to propose QUAG-attention, a replacement of self-attention in fusion module that calculates similarities on al-

ready short-circuited sequences. That is, QUAG-attention intrinsically induces the models to infer using sub-optimal representations.

Let us consider the case such that the performance of $M$ under video short-circuiting operation is comparable to its performance without any perturbation. If the input of $M$ is $\boldsymbol{X}_0 = [\boldsymbol{F}_{\mathcal{V}}(\mathcal{V}) \parallel \boldsymbol{F}_{\mathcal{T}}(\mathcal{T})]$, then during token-mixing we effectively average and scale the components in the upper-partition ($[1, \cdots, l_{\mathcal{V}}] \times d$) of the value matrix in all the fusion blocks. This can be efficiently approximated by replacing the entire upper block with a single row-wise average token using $\mathcal{R}$ before projecting to key and value domains. Note that the query remains unchanged. Similar to QUAG, we perform no finetuning and only modify the calculation of self-attention.

We can generalize it to present new variants of self-attention: collectively known as QUAG-attention. QUAG-attention operates by consistently averaging the corresponding modality blocks within the input of each fusion block. The averaging process occurs prior to the transformation of the input into keys and values. Depending on the sub-optimalities in representation, QUAG-attention can be applied to only text, video or both the modalities. It reduces the number of keys and values tokens from ($l_{\mathcal{V}} + l_{\mathcal{T}}$) to either ($l_{\mathcal{T}} + 1$) (text-average), ($l_{\mathcal{V}} + 1$) (video-average) or 2 (text-video-average).

The number of tokens in video and text modalities are generally different. However, due to block averaging, QUAG-attention reduces the effective number of tokens of the modality in key and value domains to one. The token-length mismatch would interfere with softmax operation in attention. Hence, we scale the components of dot-product similarity scores of the averaged keys by the logarithm of the number constituting tokens (that is, the original number of tokens in the block). This is similar to proportional attention used by Bolya et al. (2023) for token-merging.

## 2.5 EXPERIMENTAL SETTING

**Models and Datasets**: We evaluate QUAG and QUAG-attention on JustAsk (Yang et al., 2021a) and FrozenBiLM (Yang et al., 2022b) models. We evalaute it on the following datasets *(i)* **ActivityNet-QA** (Yu et al., 2019): contains 58K open-ended questions on 5.8K sampled videos from ActivityNet *(ii)* **MSRVTT-QA** (Xu et al., 2017): contains 244K open-ended questions on 10K MSRVTT videos *(iii)* **NeXT-QA** (Xiao et al., 2021): contains 47K 5-way multiple choice questions with one-correct answer from 5.4K videos. We also report results on the **ATP-Hard** subset of NeXT-QA (Buch et al., 2022) that contains a higher concentration of temporally challenging data requiring multi-frame understanding. **Implementation Details**: All our experiments were performed on 4 NVIDIA A5000 GPUs. We use the official open-source code of the models on GitHub and modify only the self-attention modules. We use the official evaluation code and checkpoints. For NeXT-QA, we use the official dataset and finetune the models with the default parameters. More details in Appendix A.2.3.

## 2.6 ANALYSIS

The results are shown in Table 1. For comparison to the unperturbed model, we specify the baseline, language-only (without video input) and video-only (without text input) accuracies. The high performance in language-only setting relative to the baseline is indicative of strong unimodal bias towards language. However, these metrics do not provide any information about the exact nature and degree of the sub-optimal representations learned by the models, hence we use QUAG.

The performance of FrozenBiLM on ActivityNet-QA and MSRVTT-QA drops by over 10% (43.6% to 32.3%; 46.6% to 32.8%) with crossmodal short-circuiting, and by 40% with both unimodal (43.6% to 2.4%; 46.6% to 1.0%) and text short-circuiting (43.6% to 1.4%; 46.6% to 1.0%). Furthermore, the drop is less than 1% under video short-circuiting (43.6% to 43.1%; 46.6% to 45.7%). However, for NeXT-QA and ATP-Hard, the performance of FrozenBiLM drops to chance level (20%) under text and unimodal short-circuiting operations but hardly drops with video and text short-circuiting. Parallelly, the performance of JustAsk model does not drop by more than 1% for any of the datasets under any short-circuting operation.

This means that FrozenBiLM consistently does not rely on the core features of the video modality and has a strong reliance on text-modality. Further, for NeXT-QA and ATP-Hard, the model does not leverage any crossmodal interactions. However, for ActivityNet-QA and MSRVTT-QA, it leverages

Table 1: Short-circuit (SC) and QUAG-attention accuracies for JustAsk and FrozenBiLM models on ActivityNet-QA (A-QA), MSRVTT-QA (M-QA), NeXT-QA (N-QA) and ATP-Hard (ATP-H) datasets (*video-average for FrozenBiLM and video-text-average for JustAsk; [†] percentage decrease in the number of multiplication operations due to QUAG-attention).

| | FrozenBiLM | | | | JustAsk | | | |
|---|---|---|---|---|---|---|---|---|
| | A-QA | M-QA | N-QA | ATP-H | A-QA | M-QA | N-QA | ATP-H |
| Baseline | 43.6 | 46.6 | 55.8 | 55.7 | 38.7 | 41.8 | 53.8 | 44.0 |
| Language-only | 32.2 | 33.2 | 55.7 | 55.8 | 28.2 | 29.9 | 42.2 | 42.0 |
| Video-only | 0.1 | 0.0 | 20.2 | 20.1 | 2.6 | 6.7 | 39.1 | 23.0 |
| SC: unimodal | 2.4 | 1.0 | 19.8 | 21.4 | 38.5 | 41.5 | 53.6 | 43.6 |
| SC: crossmodal | 32.3 | 32.8 | 56.0 | 55.6 | 38.3 | 41.3 | 53.5 | 44.3 |
| SC: video | 43.1 | 45.7 | 55.8 | 55.7 | 38.2 | 41.3 | 53.4 | 44.3 |
| SC: text | 1.4 | 1.0 | 20.5 | 21.1 | 38.6 | 41.5 | 53.7 | 43.6 |
| QUAG-atten* | 43.0 | 45.8 | 55.6 | 55.9 | 38.0 | 41.0 | 53.5 | 44.1 |
| $\Delta$MulOps[†] | **13.6%** | | | | **68.0%** | | | |

some crossmodal interactions (video (query) and text (key) only). On the other hand, JustAsk model does not learn to fuse the modalities across the datasets and relies largely on the text-modality. Note that while the relative performance drop in the *classical* language-only and video-only settings for JustAsk and FrozenBiLM models on ActivityNet-QA and MSRVTT-QA is similar, QUAG points out the differences in their sub-optimal representations.

We use the results from QUAG to apply QUAG-attention on FrozenBiLM and JustAsk that reduce the number of multiplication operations by **13.6%** and **68.0%** respectively, for a less than 1% drop in performance consistently for all the datasets. However, this raises serious concerns because models can learn to *hack* their way around the accuracy metrics for leveraging shortcuts. The supposedly multimodal datasets contain biases and the evaluation metrics do not penalize shortcut learning and provide a false confidence about the abilities of the model. This raises the follow-up question – **How can we confidently benchmark multimodal understanding in VideoQA models?**

## 3    DOES MULTIMODAL SUB-OPTIMALITY STEMS FROM DATASET BIASES?

Sub-optimality in model representations and shortcut learning can stem from a combination of facets like dataset biases, model architecture (Jelassi et al., 2022), optimization method (Gunasekar et al., 2018), learning paradigm (Liu et al., 2022) etc. Hence, to ablate the effect of dataset biases we curate CLAVI, a diagnostic dataset with temporal counterfactuals in questions and videos that necessitates joint multimodal understanding and penalizes simple shortcut learning. CLAVI is not positioned to replace existing datasets but rather to supplement them, enhancing the understanding of VideoQA models. We finetune the VideoQA models on CLAVI with the prescribed model architecture and training recipe to study and diagnose the representational prowess of the pretrained models.

### 3.1    CLAVI: DIAGNOSING THROUGH COUNTERFACTUALS

CLAVI consists of 6,018 videos and 114,342 questions (72,770 train and 41,572 test). It contains simple yes-no questions to probe the absolute temporal location of a single action (beginning/end) or the occurrence sequence for a pair of non-overlapping actions (before/after). CLAVI allows for systematic benchmarking and diagnosis of joint multimodal understanding through the lens of balanced video and question temporal counterfactuals. We use question templates to automatically curate the question-answer pairs from the temporal grounding annotations of Charades-STA (Gao et al., 2017). To create temporal counterfactuals in the question domain, we replace *before* with *after* and *beginning* with *end* and vice versa. Further, we create temporal counterfactuals in the video domain by swapping only the action-segments in the video as shown in Figure 2. We exhaustively consider all the compositions of temporal counterfactuals in video and question domains to create balanced counterfactuals instances for systematic assessment of multimodal understanding in videos.

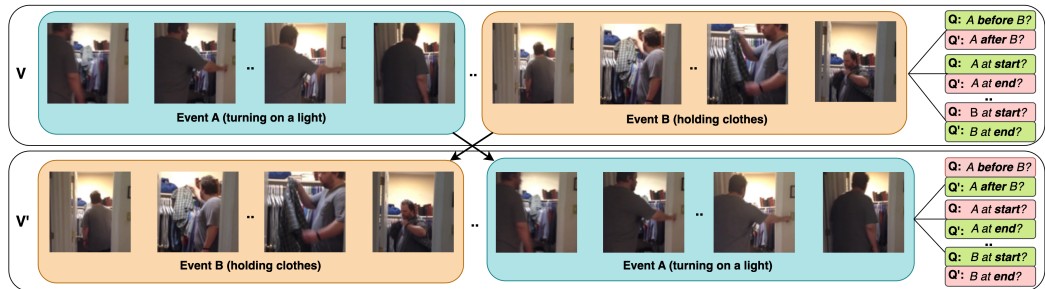

Figure 2: Illustrative example of the creation of CLAVI. In the original video (V), the action "turning on a light" (Event A; blue pane) follows "holding clothes" (Event B; brown pane). To create a counterfactual video (V'), we swap the action segments without manipulating the segment separating them. The questions (Q), along with their counterfactual (Q'), are curated for each of the videos. Note that the color of the question panel reflects the correct answer (green for "yes", pink for "no"). We provide the list of questions in Table 2.

Table 2: List of questions and their counterfactuals in CLAVI for the illustrative example in Fig. 2. For brevity, we present 4 (out of 8) negative control (NC) questions for BA type; comprehensive list in Appendix A.3.3.

| Question (Q) | Counterfactual Question (Q') |
|---|---|
| **Existence (E) type** | |
| Was someone turning on light? | |
| Was someone holding clothes? | |
| **Existence (E) type – (NC)** | |
| Was someone washing mirror? | |
| **Beginning/End (BE) type** | |
| Was the person turning on light at the **beginning**? | Was the person turning on light at the **end**? |
| Was the person holding clothes at the **end**? | Was the person holding clothes at the **beginning**? |
| **Before/After (BA) type** | |
| Did turning on light happen **before** holding clothes? | Did turning on light happen **after** holding clothes? |
| Did holding clothes happen **after** turning on a light? | Did holding clothes happen **before** turning on light? |
| **Before/After (BA) type – (NC)** | |
| Did turning on light happen **before** washing mirror? | Did turning on a light happen **after** washing mirror? |
| Did holding clothes happen **after** washing mirror? | Did holding clothes happen **before** washing mirror? |

We briefly explain the design principle of CLAVI. We choose temporal sequence counterfactuals to benchmark joint multimodal understanding because it requires **unimodal understanding** within the modalities (sensitive to the sequence of *(i)* frames in the video; *(ii)* objects, verbs and temporal phrases in the question) as well as **crossmodal understanding** (relating the sequence of actions in the video with that of the question). This also makes temporal ordering as one of the **fundamental** elements of VideoQA. Using yes-no questions with balanced negative instances allows us to have questions that are **unambiguous**, and answers that are **mutually exclusive** and **equally informative** to not be eliminated by prior biased knowledge. We deliberately maintain a simple design for question templates and answer vocabulary that excludes other abilities such as language comprehension, commonsense reasoning, and long-term memory to facilitate **isolated diagnostic analysis** of joint multimodal understanding. Also, we ensure that the dataset size is sufficiently large, as compared to the existing datasets, so that the models do not overfit (Appendix A.3.2).

Based on the temporal cue in the question, CLAVI contains three question types – **Existence (E)**, **Beginning/End (BE)** and **Before/After (BA)**. Further, we define negative control questions containing actions that do not occur in the video (that is, the answer is always "no") for E and BA types as shown in Table 2. Answering the negative control does not require understanding temporal cues in language and video. Hence, it serves the dual purpose of sanity check of learning and a baseline for learning by temporal shortcuts. We remove the bias against *beginning* and *end* by randomly

Table 3: Test performance (% accuracy) on CLAVI after finetuning

| Metric | JustAsk (Yang et al., 2021b) | FrozenBiLM (Yang et al., 2022a) | Singularity-T (Lei et al., 2023) | All-In-One+ (Wang et al., 2023a) |
|---|---|---|---|---|
| Balanced Acc | 72.2 ± 0.2 | 80.5 ± 0.1 | 76.8 ± 0.5 | 73.9 ± 0.1 |
| $CAcc_\mathcal{V}$ | 50.6 ± 0.3 | 74.0 ± 0.1 | 47.2 ± 1.1 | 49.6 ± 0.5 |
| $CAcc_\mathcal{T}$ | 50.3 ± 0.1 | 75.5 ± 0.1 | 47.0 ± 1.0 | 49.5 ± 0.3 |
| $CAcc_\mathcal{V}$-control | 98.0 ± 0.2 | 93.2 ± 0.2 | 92.7 ± 2.0 | 98.1 ± 0.5 |
| $CAcc_\mathcal{T}$-control | 98.2 ± 0.2 | 93.7 ± 0.2 | 93.5 ± 1.9 | 98.2 ± 0.7 |
| $CAcc_\mathcal{V}$-counter | **3.6 ± 0.1** | 54.1 ± 0.2 | **1.7 ± 0.2** | **1.2 ± 0.3** |
| $CAcc_\mathcal{T}$-counter | **2.4 ± 0.1** | 57.2 ± 0.2 | **0.5 ± 0.2** | **0.8 ± 0.1** |

extending the boundaries of the action-segments in the video. The detailed curation process and dataset statistics are presented in Appendix A.3.1.

We want to evaluate the sensitivity of the model to the temporal cues in language and video independently. Hence, we define consistent accuracies. Given a question, if the model predicts the answers correctly for both – the video and its corresponding counterfactual video, it is called video-consistent. Similarly, for a given video, if the model correctly answers a question and it corresponding counterfactual question, it is called text-consistent. The proportion of video and question consistent predictions are reported as **video-consistent accuracy ($CAcc_\mathcal{V}$)** and **text-consistent accuracy ($CAcc_\mathcal{T}$)** respectively. We report the consistent accuracies separately for the **control subset** (E, E-NC, and BA-NC question types) and the **counterfactual subset** (BE and BA question types). The control subset can be answered by leveraging shortcuts while answering the counterfactual subset necessitates joint multimodal understanding.

## 3.2 EXPERIMENT

We finetune and evaluate 4 models: JustAsk (Yang et al., 2021a), FrozenBiLM (Yang et al., 2022b), Singularity-Temporal (Lei et al., 2023) and All-In-One+ (Wang et al., 2023b) on CLAVI using the official finetuning instructions (Appendix A.3.5). We follow the same experimental settings as discussed in Section 2.5. To account for class imbalance in the answers, we use balanced accuracy for validation and testing. The results are summarized in Table 3. All the models have greater than 70% balanced accuracy. At first, it might give an illusion of good multimodal understanding in VideoQA models. However, the consistent accuracy metrics demystify the illusion.

Text and video consistent accuracies are greater than 90% for the control subset for all the models. This is because, unlike the counterfactual subset, the control subset does not requires coupled understanding. That is, the model can answer it correctly by simple shortcuts – irrespective of the context of the negative control action in the question and the location of the object and/or the action in the video. However, for achieving high consistent accuracies on the counterfactual subset, the model needs to jointly understand the order of the events and the temporal cues in the question along with the order of the events in the video. We get significantly lower consistent accuracies (less than 4%) for the counterfactual subset, except for FrozenBiLM. Overall, this means that the other **models are able to exploit shortcuts but unable to learn joint multimodal representations**.

How can we be sure that FrozenBiLM is not learning spurious shortcuts on CLAVI? We find that the video-average QUAG-attention on FrozenBiLM cause the $CAcc_\mathcal{T}$-counter and $CAcc_\mathcal{V}$-counter to drop to **23%** and **3.6%** respectively. That is, the performance on the counterfactual subset significantly drops under multimodal impairment. However, $CAcc_\mathcal{T}$-control and $CAcc_\mathcal{V}$-control values increase to **98.6%** and **99.2%** respectively, perhaps because QUAG-attention promotes reliance on shortcuts, and the control subset can be solved easily by shortcuts. These confirm FrozenBiLM's reliance on multimodal representations for its high performance relative to the other models.

Beyond the consistency accuracy metrics we can use CLAVI for diverse representation analyses. As an example, we present a qualitative representation sensitivity analysis for FrozenBiLM in Appendix A.3.7. We align the attention matrices for counterfactual pairs and find that the representations of correctly answered counterfactual pairs are more distinct than the wrongly answered pairs to validate joint multimodal understanding.

## 4 RELATED WORK

**Dataset Biases**: Works in NLP (Papadimitriou et al., 2022; Sinha et al., 2021), vision (Brendel & Bethge, 2019) and vision-language (Yuksekgonul et al., 2023) demonstrate that models can achieve high performance without even understanding the sequence of the embeddings. This is partly because the current benchmarks have unintended biases that could potentially be exploited by models to learn shortcuts; hence accuracy is not always a faithful metric (Pham et al., 2021; Yuksekgonul et al., 2023; Kafle & Kanan, 2017; Sevilla-Lara et al., 2021). For VideoQA, MovieQA (Tapaswi et al., 2016) and TVQA (Lei et al., 2018) datasets are biased towards plot understanding or dialogue comprehension (Winterbottom et al., 2020). Biases are not always immediately apparent; for example, Social-IQ (Zadeh et al., 2019) contains sentiment-biased annotations (Gat et al., 2021). Moreover, statistical regularities like answer length, answer frequency (Goyal et al., 2017; Agrawal et al., 2016) and co-occurrence (Dancette et al., 2021a; Manjunatha et al., 2019; Subramanian et al., 2019) introduce spurious features. Overall, these biases allow the models learn shortcuts (Geirhos et al., 2020) that circumvent multimodal reasoning (Chao et al., 2018; Ye & Kovashka, 2021). While synthetic VideoQA benchmarks such as VQuAD (Gupta et al., 2022), CLEVRER (Yi et al., 2019) have been carefully curated to mitigate many biases, they are unable to capture the intricate dynamics of the real world. Recently proposed Preception Test (Pătrăucean et al., 2023), while comprehensive, does not contain diagnostic metrics that penalize the effect of shortcut learning. We curate CLAVI by systematically augmenting real-world videos to faithfully represent the complexity of the physical world while controlling the biases to confidently evaluate multimodal temporal understanding.

**Shortcut Learning**: Tangential to the bias amelioration methods (Cadene et al., 2019; Clark et al., 2019), Lei et al. (2023) and Winterbottom et al. (2020) achieve state-of-the-art performance with simple models by leveraging VideoQA dataset shortcuts in the model. ATP (Buch et al., 2022) demonstrates single frame bias by re-training the models with an informative frame-selection module to achieve competitive performance. Perceptual Score (Gat et al., 2021) quantifies modality bias in terms of relative performance drop under modality-permutation operation. QUAG combines these ideas to evaluate the dependence of models on shortcuts for circumventing multimodal understanding in terms of performance drop under multimodal representation collapse. Unlike others, it assists in identifying sub-optimal representations in a combined model-dataset approach at test time.

**Leveraging Counterfactuals**: We share our motivation for developing CLAVI with VQA-CP (Agrawal et al., 2018): that iid train-test splits in the presence of strong priors leads to learning via shortcuts. However, rather than reducing the bias by mining new complementary image instances, CLAVI weakens prior of multimodal understanding with synthesized balanced video-question temporal hard-negatives. Concurrent to our work, Momeni et al. (2023) and Wang et al. (2023c) have employed hard-negatives for improving verb-understanding in VideoQA models. Bagad et al. (2023) stitch pairs of unrelated videos to improve the temporal understanding of video-language models. However, unlike CLAVI that uses synthesized negative video instance from the same video, stitched video dataset cannot be a robust diagnostic benchmark because the incoherent contexts can be exploited as a static bias shortcut (Choi et al., 2019).

## 5 CONCLUSION

In this work, we perform a rigorous analysis of VideoQA models, focusing on multimodal representations. We introduced QUAG, a coupled dataset-model approach, to conduct a systematic analysis of learned multimodal representations. It provides deep insights into *how* the models infer and *why* the models fail. We found that VideoQA models can learn shortcuts on seemingly multimodal datasets without truly learning to align and fuse the information both – within and between the modalities. Using this understanding, we developed QUAG-attention and exposed the sub-optimality of VideoQA models. Hence, we proposed CLAVI, a diagnostic benchmark for analyzing joint multimodal understanding in VideoQA models. With the simple task of temporal ordering we find that most of the current models are unable to jointly infer from text and video modalities. All our proposed approaches – QUAG, QUAG-attention and CLAVI are simple, compute-friendly and generic to be extended to any combination of modalities, datasets and models. Our thorough and systematic dataset-model combined representation analysis provides insights that are shrouded and misled by the standard datasets and evaluation metrics that create the illusion of joint multimodal understanding.

## 6 ETHICAL STATEMENT

Datasets in machine learning have a history of containing unintentional biases like race, gender, age along with safety and privacy concerns (Birhane & Prabhu, 2021; Peng et al., 2021; Hirota et al., 2022). We curate CLAVI from existing and popular Charades (Sigurdsson et al., 2016) dataset because it is well-studied and collected in controlled settings with consent. Further the owners of Charades ensure the anonymity and privacy of the participants. However, our approach of developing CLAVI is quite generic and can be easily extended to multiple datasets and/or modalities. Also, for automatically generated questions, we ensure to keep the question templates gender neutral.

One of the central goals of our work was to reassert the brittleness in multimodal models by presenting a combined dataset-model centric interpretable representation learning approach through QUAG and CLAVI. We hope our work galvanizes the research community further to not just blindly trust the accuracy score on benchmarks but thoroughly investigate the potential biases that are *(1)* present in the dataset and *(2)* are learned by the models.

## 7 REPRODUCIBILITY STATEMENT

We provide the code to implement QUAG and QUAG-attention in Appendix A.2.2. It can be easily integrated by replacing the self-attention module of the models. Also, QUAG and QUAG-attention are deterministic and non-parametric. Hence, the reproducibility of the results is ensured by reproducibility of the model codebase which we have thoroughly verified.

We provide the scripts to curate CLAVI from Charades along with a representative subset of the dataset in the supplementary material because of licensing restrictions. We have ensured that the dataset curation code is reproducible and it will be released on GitHub under GPL3.0 license. Following the guidelines of Gebru et al. (2021), we provide a comprehensive documentation of CLAVI with the intended use in Appendix A.3.8. For the finetuning experiments on CLAVI, all the experiments were repeated with three random seeds (0, 42 and 71) and the mean results reported with the standard error. We will release the finetuned checkpoints with the code on GitHub post acceptance notification.

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

## A    APPENDIX

### A.1    ADDITONAL RELATED WORKS

**Multimodal Fusion Interpretability and Visualization:**    Liang et al. (2022b) and Liang et al. (2023) analyze multimodal fusion interactions along the dimensions of response, information, and mechanics. The closest alignment of QUAG and CLAVI is at the interface of multimodal fusion response and mechanics. Previous works have quantified the presence or absence of specific kinds of modality interactions through the study of datasets (Dancette et al., 2021b), models (Chefer et al., 2021), projections onto simpler models (Hessel & Lee, 2020; Wörtwein et al., 2022) and visualization studies (Liang et al., 2022a; Aflalo et al., 2022; Wang et al., 2021). However, using QUAG, unlike other methods, we perform a combined dataset-model analysis without any additional parameters or finetuning.

### A.2    QUAG

#### A.2.1    ATTENTION MAP VISUALIZATION

We provide a visualization example of the attention values before and after short-circuting operations in Figure 3.

#### A.2.2    CODE

Below is the implementation of QUAG as an augmentation of the existing self-attention function. We use row-wise average and replace operation in each if-clause statements, while ignoring the padding, to ablate the effect of the quadrant.

```
1 def self_attention(inputs, mask, dim_model, l_v, l_t, quads):
2     # Inputs:
```

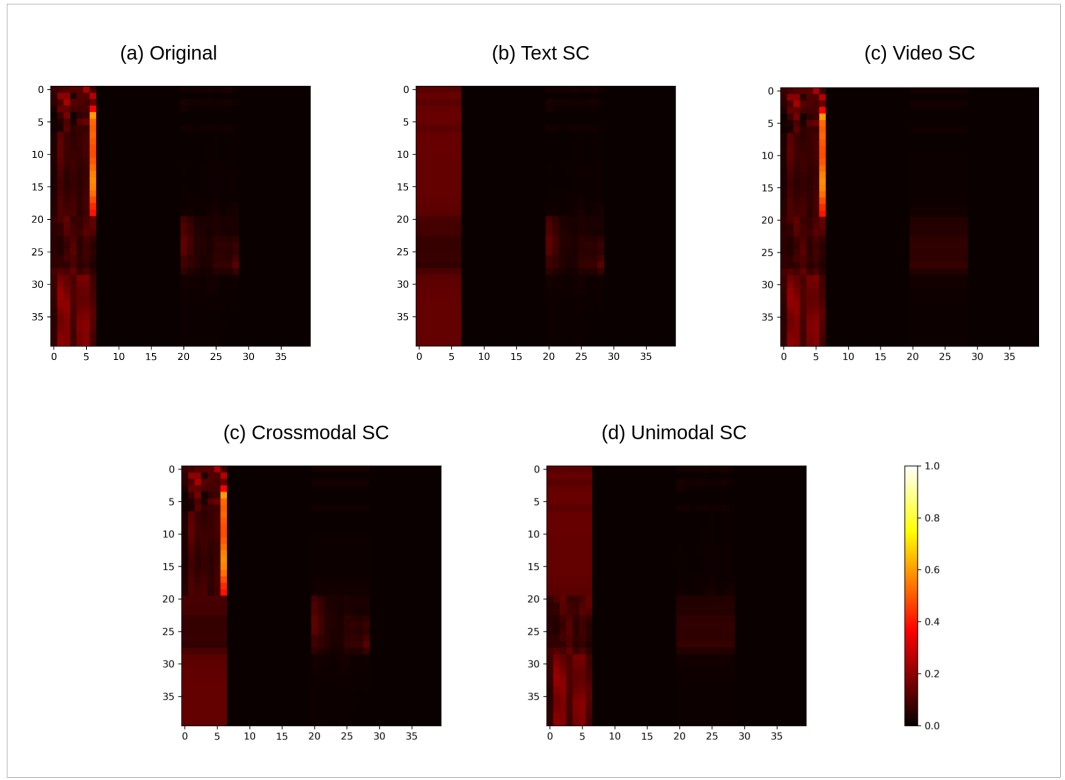

Figure 3: Visualization of the first attention head, as a heatmap, from the second layer of JustAsk model with $l_\mathcal{V} = 20$ and $l_\mathcal{T} = 20$. Note that here the text embeddings are pre-concatenated to the video embedding in the input. The lengths of the video and text tokens are 9 and 7 respectively. The text and video tokens are individually padded to length 20 each. We visualize *(a)* the original attention values and *(b)-(d)* after short-circuiting (SC) operations.

```
3     #   inputs: Tensor of shape (batch_size, sequence_length,
      dim_model)
4     #   mask: Tensor of shape (batch_'size, sequence_length)
5     #   dim_model: Dimension of the model (e.g., 512)
6     #   l_v: int    maximum length of video tokens
7     #   l_t: int    maximum length of question tokens
8     #   quads: list containing elements from {'VV', 'VT', 'TV', 'TT'}
9     query = linear_transform_query(inputs)
10    key = linear_transform_key(inputs)
11    value = linear_transform_value(inputs)
12    attention_scores = compute_attention_scores(query, key, mask)
13    apply_quag(attention_scores, mask, l_v, l_t, quads)
14    attended_output = apply_attention_scores(attention_scores, value)
15    return attended_output
16
17 def compute_attention_scores(query, key, mask):
18    scaled_dot_product = dot_product(query, key) / sqrt(dim_model)
19    attention_scores = softmax(scaled_dot_product + (1 - mask) * -1e9)
20    return attention_scores
21
22 def apply_quag(attention_scores, mask, l_v, l_t, quads):
23    if 'VV' is in quads:
24        replace_with_rowwise_average(attention_scores[:, :l_v, :l_v],
      mask[:, :l_v, :l_v])
25    if 'VT' is in quads:
26        replace_with_rowwise_average(attention_scores[:, :l_v, -l_t:],
       mask[:, :l_v, -l_t:])
```

```
27      if 'TV' is in quads:
28          replace_with_rowwise_average(attention_scores[:, -l_t:, :l_v],
        mask[:, -l_t:, :l_v])
29      if 'TT' is in quads:
30          replace_with_rowwise_average(attention_scores[:, -l_t:, -l_t
        :], mask[:, -l_t:, -l_t:])
31
32 def replace_with_rowwise_average(scores, mask):
33      rowwise_sum = sum(scores, axis=-1)
34      rowwise_mean = rowwise_sum / sum(mask, axis=-2)
35      expanded_rowwise_mean = expand_dims(rowwise_mean, axis=-1)
36      replace_elements(scores, expanded_rowwise_mean)
37
38 def apply_attention_scores(attention_scores, value):
39      attended_output = dot_product(attention_scores, value)
40      return attended_output
```

Next, we provide the code for QUAG-attention. QUAG-attention modifies the existing self-attention block in the fusion module by replacing the block with the block average. We also demonstrate the normalizing the softmax function so that the each single average sequence is representative of the constituent sequences.

```
1 def quag_attention(inputs, mask, dim_model, l_v, l_t, type):
2      # Inputs:
3      #   inputs: Tensor of shape (batch_size, sequence_length,
        dim_model)
4      #   mask: Tensor of shape (batch_size, sequence_length)
5      #   dim_model: Dimension of the model (e.g., 512)
6      #   l_v: int     maximum length of video tokens
7      #   l_t: int     maximum length of question tokens
8      #   type: one of 'text', 'video', 'text-video'
9      query = linear_transform_query(inputs)
10     avg_input = compute_avg_input(inputs, l_v, l_t, type)
11     key = linear_transform_key(avg_input)
12     value = linear_transform_value(avg_input)
13     mask = apply_mask(mask, l_v, l_t, type)
14     scaled_dot_product = compute_scaled_dot_product(query, key,
        dim_model, mask)
15     attention_scores = softmax(scaled_dot_product)
16     attended_output = apply_attention_scores(attention_scores, value)
17     return attended_output
18
19 def compute_avg_input(inputs, l_v, l_t, type):
20     if type == "video":
21         avg_upper_block = sum(inputs[:, :l_v, :], axis=-2)
22         avg_upper_block = expand_dims(avg_upper_block, axis=1)
23         avg_input = concatenate((avg_upper_block, inputs[:, :-l_t, :])
        , axis=1)
24     elif type == "text":
25         avg_lower_block = sum(inputs[:, :-l_t, :], axis=-2)
26         avg_lower_block = expand_dims(avg_lower_block, axis=1)
27         avg_input = concatenate((inputs[:, :l_v, :], avg_lower_block),
         axis=1)
28     elif type == "text-video":
29         avg_upper_block = sum(inputs[:, :l_v, :], axis=-2)
30         avg_upper_block = expand_dims(avg_upper_block, axis=1)
31         avg_lower_block = sum(inputs[:, :-l_t, :], axis=-2)
32         avg_lower_block = expand_dims(avg_lower_block, axis=1)
33         avg_input = concatenate((avg_upper_block, avg_lower_block),
        axis=1)
34     return avg_input
35
36 def apply_mask(mask, l_v, l_t, type):
37     mask = expand_dims(mask, axis=-1)
```

Table 4: Fine-grained performance of JustAsk on ActivityNet-QA

| Config | Motion | Spatial | Temp | Y/N | Color | Obj | Loc | Num | Other |
|---|---|---|---|---|---|---|---|---|---|
| Baseline | 30.6 | 19.9 | 4.9 | 64.2 | 34.7 | 26.7 | 35.5 | 48.9 | 36.8 |
| Lang-only | 1.4 | 9.1 | 4.3 | 51.8 | 28.7 | 23.0 | 16.6 | 46.9 | 29.1 |
| Vid-only | 20.3 | 0.9 | 1.8 | 0.0 | 0.0 | 1.6 | 1.3 | 0.0 | 0.7 |
| SC: unimodal | 30.1 | 19.1 | 4.9 | 63.9 | 33.6 | 26.4 | 36.8 | 48.4 | 37.0 |
| SC: crossmodal | 28,0 | 18.9 | 4.8 | 64.7 | 34.7 | 25.8 | 35.5 | 48.5 | 36.4 |
| SC: text | 30.4 | 19.3 | 5.0 | 64.1 | 34.0 | 26.4 | 35.5 | 46.7 | 37.2 |
| SC: video | 28.6 | 18.8 | 4.5 | 64.3 | 34.6 | 25.5 | 35.5 | 48.4 | 36.1 |
| QUAG-attention | 28.1 | 18.5 | 4.9 | 64.1 | 33.6 | 25.2 | 34.7 | 48.0 | 36.6 |

```
38      mask = tile(mask, [1, 1, sequence_length])
39
40      if "video" in type:
41          video_length = sum(mask[:, :l_v, 0], axis=1)
42          video_length = expand_dims(video_length, axis=-1)
43          scaled_dot_product[:, :, 0] = scaled_dot_product[:, :, 0] *
        log(video_length)
44          upper_mask = ones(mask.shape[0], mask.shape[1], 1)
45          mask = concatenate((upper_mask, mask[:, :, l_v:]), axis=-1)
46
47      if "text" in type:
48          text_length = sum(mask[:, :-l_t, 0], axis=1)
49          text_length = expand_dims(text_length, axis=-1)
50          scaled_dot_product[:, :, -1] = scaled_dot_product[:, :, -1] *
        log(text_length)
51          lower_mask = ones(mask.shape[0], mask.shape[1], 1)
52          mask = concatenate((mask[:, :, :-l_t], lower_mask), axis=-1)
53
54      return mask
55
56  def compute_scaled_dot_product(query, key, dim_model, mask):
57      scaled_dot_product = dot_product(query, key) / sqrt(dim_model)
58      return scaled_dot_product
59
60  def apply_attention_scores(attention_scores, value):
61      attended_output = dot_product(attention_scores, value)
62      return attended_output
```

### A.2.3    EXPERIMENT DETAILS

As mentioned in the main manuscript, we use the official checkpoints and code of JustAsk [website] and FrozenBiLM [website]. For all the experiments with JustAsk, we use the checkpoints of the model pretrained on HowToVQA69M and WebVidVQA3M. For FrozenBiLM, we use the WebVid10M-pretrained checkpoint for all our experiments. Since QUAG operates at inference time, we do not need to perform any training. Since the model owners do not report results on NeXT-QA, we finetune the models with the official recipe to achieve performance similar to that independently reported by others Xiao et al. (2022). While FrozenBiLM can also take subtitles as the input, for fair comparison, we do not pass it in any of the experiments. We provide the hardware details in the main manuscript.

### A.2.4    FINEGRAINED ACCURACIES

### A.2.5    JUSTASK MODEL

We present the fine-grained performance of JustAsk on the discussed datasets in Tables 4, 5, 6, and 7

Table 5: Fine-grained performance of JustAsk on MSRVTT-QA

| Config | What | How | Color | Where | Who | When |
|---|---|---|---|---|---|---|
| Baseline | 35.8 | 83.7 | 51.7 | 39.4 | 51.3 | 82.3 |
| Lang-only | 24.3 | 83.3 | 43.4 | 30.5 | 37.1 | 72.3 |
| Vid-only | 8.5 | 0.0 | 3.5 | 0.4 | 3.0 | 10.1 |
| SC: unimodal | 35.6 | 83.3 | 51.8 | 39.8 | 50.8 | 82.3 |
| SC: crossmodal | 35.35 | 83.75 | 51.98 | 39.8 | 50.8 | 81.8 |
| SC: text | 35.7 | 83.2 | 51.8 | 39.0 | 50.8 | 82.1 |
| SC: video | 35.4 | 83.8 | 51.8 | 39.8 | 50.7 | 81.6 |
| QUAG-attention | 35.1 | 83.5 | 51.1 | 38.6 | 50.2 | 82.1 |

Table 6: Fine-grained performance of JustAsk on NeXT-QA

| Config | Causal | Temporal | Descriptive |
|---|---|---|---|
| Baseline | 50.8 | 52.8 | 65.0 |
| Lang-only | 39.5 | 44.3 | 47.1 |
| Vid-only | 39.2 | 37.9 | 44.0 |
| SC: unimodal | 50.5 | 52.5 | 65.3 |
| SC: crossmodal | 50.8 | 51.8 | 65.0 |
| SC: text | 50.7 | 52.7 | 65.0 |
| SC: video | 50.7 | 52.1 | 65.0 |
| QUAG-attention | 50.8 | 52.0 | 65.1 |

Table 7: Fine-grained performance of JustAsk on ATP-Hard subset of NeXT-QA

| Config | Causal | Temporal |
|---|---|---|
| Baseline | 44.4 | 43.4 |
| Lang-only | 41.2 | 43.1 |
| Vid-only | 23.5 | 22.3 |
| SC: unimodal | 43.2 | 43.3 |
| SC: crossmodal | 44.2 | 44.4 |
| SC: text | 43.7 | 43.4 |
| SC: video | 44.3 | 44.4 |
| QUAG-attention | 44.2 | 43.9 |

### A.2.6 FROZENBiLM MODEL

We present the fine-grained performance of FrozenBiLM on the discussed datasets in Tables 8, 9, 10, and 11

### A.2.7 ADDITIONAL RESULTS

We evaluated QUAG on All-in-one model and find that, as the authors claim, the model utilizes both – unimodal and cross-modal modality interactions. The results are summarized in Table A.2.7.

### A.2.8 PROGRESSIVE QUAG

We apply QUAG progressively to the first $n$ fusion layers of FrozenBiLM to find their relative importance in multimodality. We apply it in the step size of 4 (24 fusion blocks in total) and report the results for ActivityNet-QA (Figure 4) and MSRVTT-QA (Figure 5) Datasets. We consistently find that the first 8 layers are the most important in unimodal interaction and text interactions (and cross-modal interactions as well for ActivityNet-QA).

Table 8: Fine-grained performance of FrozenBiLM on ActivityNet-QA

| Config | Motion | Spatial | Temp | Y/N | Color | Obj | Loc | Num | Other |
|---|---|---|---|---|---|---|---|---|---|
| Baseline | 30.1 | 22.5 | 6.4 | 75.6 | 34.6 | 27.7 | 37.1 | 55.8 | 41.6 |
| Lang-only | 2.6 | 10.5 | 4.8 | 63.3 | 32.3 | 23.9 | 16.6 | 44.7 | 31.6 |
| Vid-only | 0.0 | 0.0 | 0.5 | 0.0 | 0.0 | 0.0 | 0.0 | 0.0 | 0.0 |
| SC: unimodal | 0.0 | 0.1 | 0.1 | 8.3 | 0.0 | 0.0 | 0.0 | 1.3 | 0.5 |
| SC: crossmodal | 1.8 | 11.1 | 3.88 | 64.5 | 32.7 | 21.7 | 16.8 | 46.0 | 32.1 |
| SC: text | 0.0 | 0.1 | 0.1 | 4.4 | 0.1 | 0.3 | 0.0 | 1.2 | 0.3 |
| SC: video | 28.8 | 21.8 | 6.5 | 75.1 | 34.3 | 29.3 | 36.0 | 55.3 | 41.0 |
| QUAG-attention | 28.9 | 22.3 | 6.0 | 74.4 | 35.0 | 27.3 | 37.3 | 54.1 | 41.1 |

Table 9: Fine-grained performance of FrozenBiLM on MSRVTT-QA

| Config | What | How | Color | Where | Who | When |
|---|---|---|---|---|---|---|
| Baseline | 40.5 | 87.2 | 57.9 | 41.5 | 56.6 | 81.4 |
| Lang-only | 27.3 | 83.6 | 50.0 | 35.8 | 41.2 | 77.6 |
| Vid-only | 0.0 | 0.0 | 0.0 | 0.0 | 0.0 | 0.0 |
| SC: unimodal | 0.7 | 0.0 | 1.2 | 0.8 | 1.8 | 0.2 |
| SC: crossmodal | 27.1 | 83.4 | 50.9 | 32.9 | 41.1 | 66.3 |
| SC: text | 0.3 | 0.0 | 0.8 | 0.0 | 2.8 | 0.0 |
| SC: video | 39.8 | 85.5 | 58.8 | 41.9 | 55.4 | 80.9 |
| QUAG-attention | 39.9 | 86.2 | 58.1 | 42.7 | 55.2 | 81.1 |

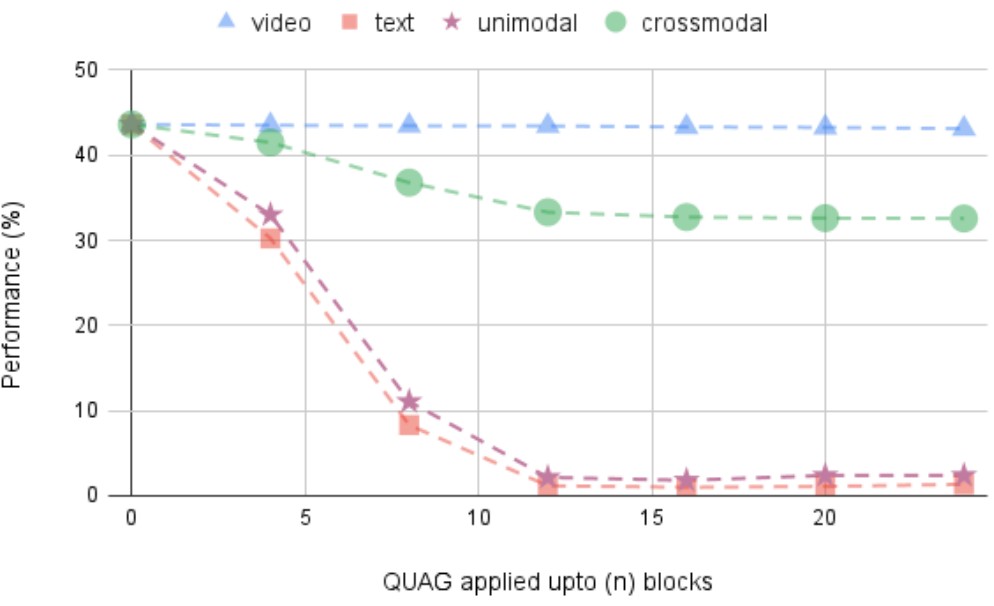

Figure 4: Result of progressive application of QUAG on FrozenBiLM model on ActivityNet-QA dataset

Table 10: Fine-grained performance of FrozenBiLM on NeXT-QA

| Config | Causal | Temporal | Descriptive |
|---|---|---|---|
| Baseline | 56.0 | 56.1 | 54.5 |
| Lang-only | 55.9 | 56.1 | 54.2 |
| Vid-only | 20.7 | 19.1 | 20.9 |
| SC: unimodal | 19.7 | 21.1 | 17.3 |
| SC: crossmodal | 56.1 | 56.5 | 54.3 |
| SC: text | 20.0 | 21.6 | 19.9 |
| SC: video | 56.1 | 56.1 | 54.5 |
| QUAG-attention | 55.9 | 55.8 | 54.1 |

Table 11: Fine-grained performance of FrozenBiLM on ATH-Hard subset of NeXT-QA

| Config | Causal | Temporal |
|---|---|---|
| Baseline | 55.2 | 56.3 |
| Lang-only | 55.5 | 56.2 |
| Vid-only | 20.0 | 20.1 |
| SC: unimodal | 20.7 | 22.5 |
| SC: crossmodal | 54.9 | 56.6 |
| SC: text | 20.2 | 22.3 |
| SC: video | 55.3 | 56.3 |
| QUAG-attention | 55.3 | 56.7 |

Table 12: Short-circuit (SC) results for All-in-one+ model on ActivityNet-QA (A-QA), and MSRVTT-QA (M-QA) datasets.

| | All-in-one+ | |
|---|---|---|
| | ActivityNet-QA | MSRVTT-QA |
| Acc | 41.9 | 43.1 |
| text_only | 23.5 | 20.8 |
| vid_only | 14.2 | 4.2 |
| SC: unimodal | 11.4 | 3.8 |
| SC: crossmodal | 20.6 | 27.6 |
| SC: video | 19.2 | 12.7 |
| SC: text | 5.6 | 7.3 |

## A.3 CLAVI

### A.3.1 DATASET CREATION

We curate CLAVI by leveraging Charades-STA (https://prior.allenai.org/projects/data/charades/license.txt) (Gao et al., 2017), containing 9,848 videos of humans performing actions based on a short script written by composing predefined vocabulary that describe multiple daily actions. The videos are annotated with the start and end times of each action. The action category, the start, and the end of each action segment are referred to as the *action tuple*. Each video may contain more than two action tuples. We select pairs of action tuples based on the uniqueness of the action category and complete exclusivity (that is no overlap between the occurrence of the actions). In a given selected pair of action tuples, the two actions along with the inter-action region constitute the video segment. We ensure that the two action categories in the pair are distinct. Additionally, to address temporal boundary ambiguities in the annotations, we filter out segments where either of the selected action classes occurs in close proximity to the segment boundaries

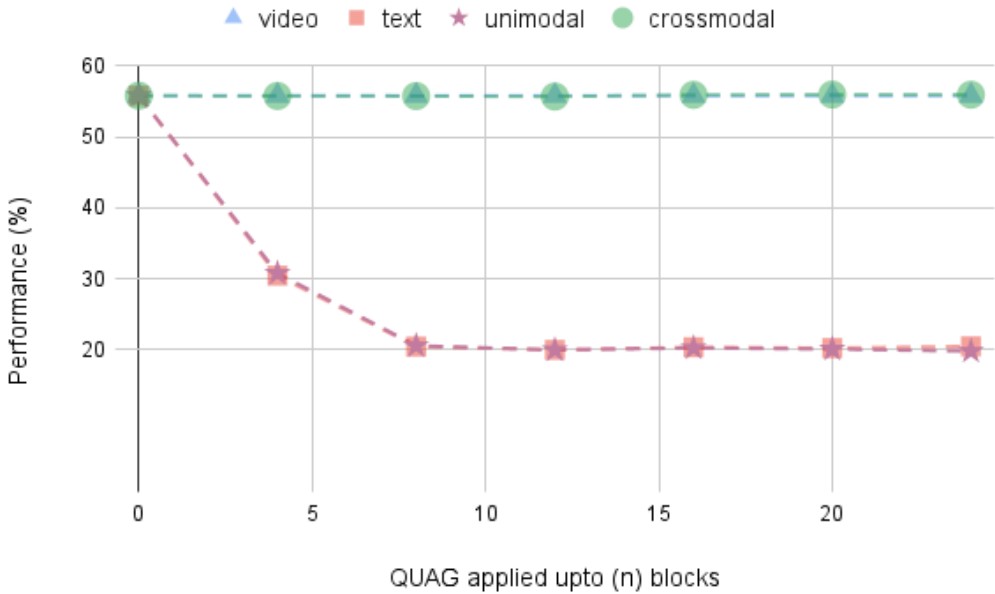

Figure 5: Result of progressive application of QUAG on FrozenBiLM model on NeXT-QA dataset

We also extend the boundaries of the two actions in the pair. We define two boundary extensions – out-extension and in-extension. The out-extension encompasses regions that are not a part of the selected segment but extend outwards in both directions into the original video. Similarly, in-extension extends inwards into the inter-action segment. To avoid temporal position bias (Hao et al., 2022; Otani et al., 2020), the lengths of the extension boundaries are selected randomly. However, since inter-action separation can affect their recognition (Bagad et al., 2023), we constraint the inter-action separation in the original and the corresponding negative video to be the same. That is, the sum of out-extension boundaries is always equal to the sum of in-extension boundaries.

We trim each boundary-extended contiguous segment from the original video to curate a positive video instance. To create the counterfactual video, we swap the boundary-extended action regions as shown in Figure 2. Note that the region between the boundary-extended actions is unaffected. Swapping operation preserves the actions but only alters their chronology, and can be applied independently to question negatives (unlike manipulations like video reversal (Wang et al., 2023c)). This independence provides fine-grained control to create a balanced benchmark for comprehensive analysis.

We create three types of questions using pre-defined templates and action-class annotations:

1) **Existence (E) type**: The E-type questions for both the action classes follow the template *"Was someone ⟨A⟩?"*, where ⟨A⟩ is one of two action classes in video. We use it as a positive control to verify if the model is able to correctly recognize the action classes. We use the exact same question for negative video instance as well, totalling to 4 control (questions, video, answer) instances for a Charades-extracted video segment.

2) **Beginning/End (BE) type**: BE type questions the absolute location of the action in the video. The question is of the form, *"Was the person ⟨A⟩ at the {beginning/end}?"* where ⟨A⟩ is one of two action classes in the video, and we select one of *beginning* and *end*. Hence, for a given video and its negative, we have, in total, 8 instances of BE (questions, video, answer) tuples combined. Note that the answer for a given BE question is complemented in the negative video.

3) **Before/After (BA) type**: BA type comprises of questions on the relative order of occurrence of actions. The question is of the form *"Did ⟨A1⟩ happen {after/before} ⟨A2⟩?"*, where ⟨A1⟩ and ⟨A2⟩ are the selected action classes. We consider all the permutations of action classes. Hence, we have a

total of 8 instances of BA type (questions, video, answer) tuples per extracted video. Similar to BE type, the answer is complemented in the negative video.

Further, we add negative controls for E and BA type questions. A negative control action is an action that does not occur in the video. Since we want to probe only for temporal understanding, we keep the negative control action-class easy to detect by randomly selecting an action-class that does not contain any of the objects or actions in the original video. Hence, answering the negative control does not require understanding temporal cues in language and video and can be answered by object elimination. It serves the dual purpose of sanity check of learning and a baseline for learning by temporal shortcuts. The answer of negative control questions is always false. This adds two E type and sixteen BA type negative control questions for the video and its negative combined. Hence, including the negative control questions, each video in CLAVI is associated with 19 questions: 2 E, 4 BE, 4 BA, 1 E negative control and 8 BA negative controls. The ratio of "yes":"no" answers is 6:13.

### A.3.2 COMPARISON WITH EXISTING DATASETS

We provide a comparison of size of CLAVI with established VideoQA datasets in Table 13.

Table 13: Comparison of CLAVI with other other VideoQA datasets sorted in the reverse order of recency.

| Dataset | Number of (V,Q,A) samples |
|---|---|
| MSRVTT-QA (Xu et al., 2017) | 243K |
| ActivityNet-QA (Yu et al., 2019) | 58K |
| Social-IQ QA (Zadeh et al., 2019) | 7.5K |
| NeXT-QA (Xiao et al., 2021) | 52K |
| iVQA (Yang et al., 2021b) | 10K |
| STAR (Wu et al., 2021) | 60K |
| EgoTaskQA (Jia et al., 2022) | 40K |
| FIBER (Castro et al., 2022) | 28K |
| NewsQA (Jahagirdar et al., 2023) | 8.6K |
| **CLAVI (Ours)** | **114K** |

### A.3.3 COMPREHENSIVE LIST OF QUESTIONS

We provide a comprehensive list of the questions for the example presented in Fig 2 of the main paper. We define the actions as: **A**: *turning on light* **B**: *holding clothes* **C**: *washing mirror*, where action A occurs before action B in the original video and action C does not occur anywhere in the original video.

Enlisted below are the questions and its negatives (Q and Q' respectively) for the video (V) (that is event A occurs after event B):Note that the color of the panel is representative of the answer of the question (red: "no", green: "yes").

**E-Type**:

> **Q :** Was someone turning on light?

> **Q :** Was someone holding clothes?

**E-Type (negative control)**:

> **Q :** Was someone washing mirror?

**BE-Type**

> **Q :** Was the person turning on light at the **beginning**?

> **Q':** Was the person turning on light at the **end**?

**Q :** Was the person holding clothes at the **end**?

**Q':** Was the person holding clothes at the **beginning**?

**BA-Type**

**Q :** Did turning on light happen **before** holding clothes?

**Q':** Did turning on light happen **after** holding clothes?

**Q :** Did holding clothes happen **after** turning on light?

**Q':** Did holding clothes happen **before** turning on light?

**BA-Type (negative-control)**

**Q':** Did washing mirror happen **before** turning on light?

**Q':** Did washing mirror happen **after** turning on light?

**Q':** Did turning on light happen **before** washing mirror?

**Q':** Did turning on light happen **after** washing mirror?

**Q':** Did washing mirror happen **before** holding clothes?

**Q':** Did washing mirror happen **after** holding clothes?

**Q':** Did holding clothes happen **before** washing mirror?

**Q':** Did holding clothes happen **after** washing mirror?

Enlisted below are the questions and its negatives (Q and Q' respectively) for the negative video instance (V') (that is event B occurs after event A).

**E-Type**:

**Q :** Was someone turning on light?

**Q :** Was someone holding clothes?

**E-Type (negative control)**:

**Q :** Was someone washing mirror?

**BE-Type**

**Q :** Was the person turning on light at the **beginning**?

**Q':** Was the person turning on light at the **end**?

**Q :** Was the person holding clothes at the **end**?

**Q':** Was the person holding clothes at the **beginning**?

**BA-Type**

**Q :** Did turning on light happen **before** holding clothes?

**Q':** Did turning on light happen **after** holding clothes?

**Q :** Did holding clothes happen **after** turning on light?

**Q':** Did holding clothes happen **before** turning on light?

**BA-Type (negative-control)**

**Q':** Did washing mirror happen **before** turning on light?

**Q':** Did washing mirror happen **after** turning on light?

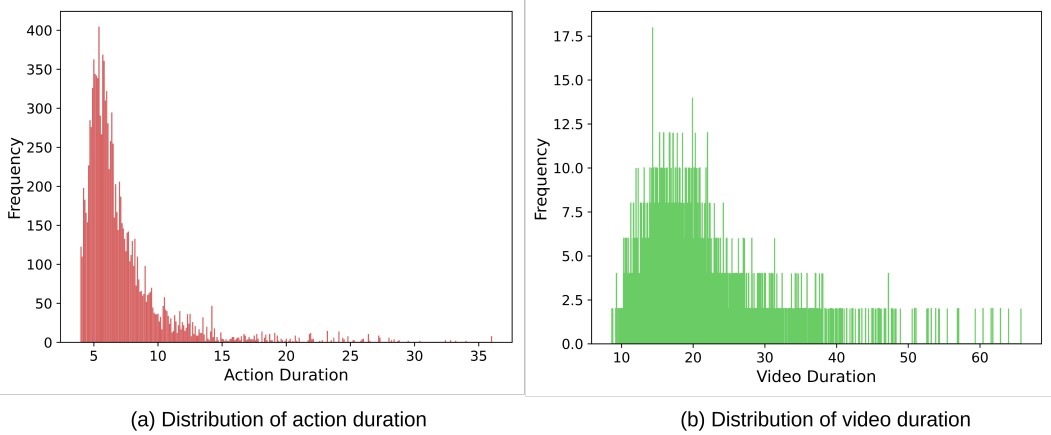

(a) Distribution of action duration          (b) Distribution of video duration

Figure 6: Distribution of length of (a) action and (b) video durations

| **Q':** Did turning on light happen **before** washing mirror? |
|---|
| **Q':** Did turning on light happen **after** washing mirror? |

| **Q':** Did washing mirror happen **before** holding clothes? |
|---|
| **Q':** Did washing mirror happen **after** holding clothes? |

| **Q':** Did holding clothes happen **before** washing mirror? |
|---|
| **Q':** Did holding clothes happen **after** washing mirror? |

### A.3.4 DATASET METRICS

The duration of individual action in CLAVI lies in the range [4.0 sec, 36.0 sec]; the average length of action is **7.7 ± 3.42** sec. The average video length is **19.95 ± 7.34** secs and the range is [8.67 sec, 65.73 sec]. We plot the distribution of the action and video durations in Fig. 6.

CLAVI consists of **141** unique action classes. Each action class is composed of noun (objects) and verb. There are **37** unique noun classes and **28** unique verb classes. We show the frequency distributions of action, verb and noun classes in Fig 7.

### A.3.5 EXPERIMENT DETAILS

As mentioned in the main manuscript, we use the official checkpoints, finetuning code and hyperparameters of JustAsk [website], FrozenBiLM [website] , Singularity-Temporal [website], and All-in-one+ [website]. For JustAsk, we use the checkpoint of the model pretrained on HowToVQA69M and WebVidVQA3M. For FrozenBiLM, we use the WebVid10M-pretrained checkpoint. All-in-one+ is pretrained on eight datasets comprising of both images and videos (videos: Webvid, YT-Temporal-180M, HowTo100M and images: CC3M, CC12M, COCO, Visual Genome, SBU Captions). Singularity-Temporal is pretrained on a 17.28M images and video subset (images: COCO, Visual Genome, SBU Captions, CC3M, CC12M and videos: WebVid). We have depicted the finetuning details in Table 14.

### A.3.6 FINE-GRAINED ACCURACIES

In Table 15 we provide error bars for the finetuning experiments. The experiments were performed thrice on the same hardware with the same set of hyperparameters.

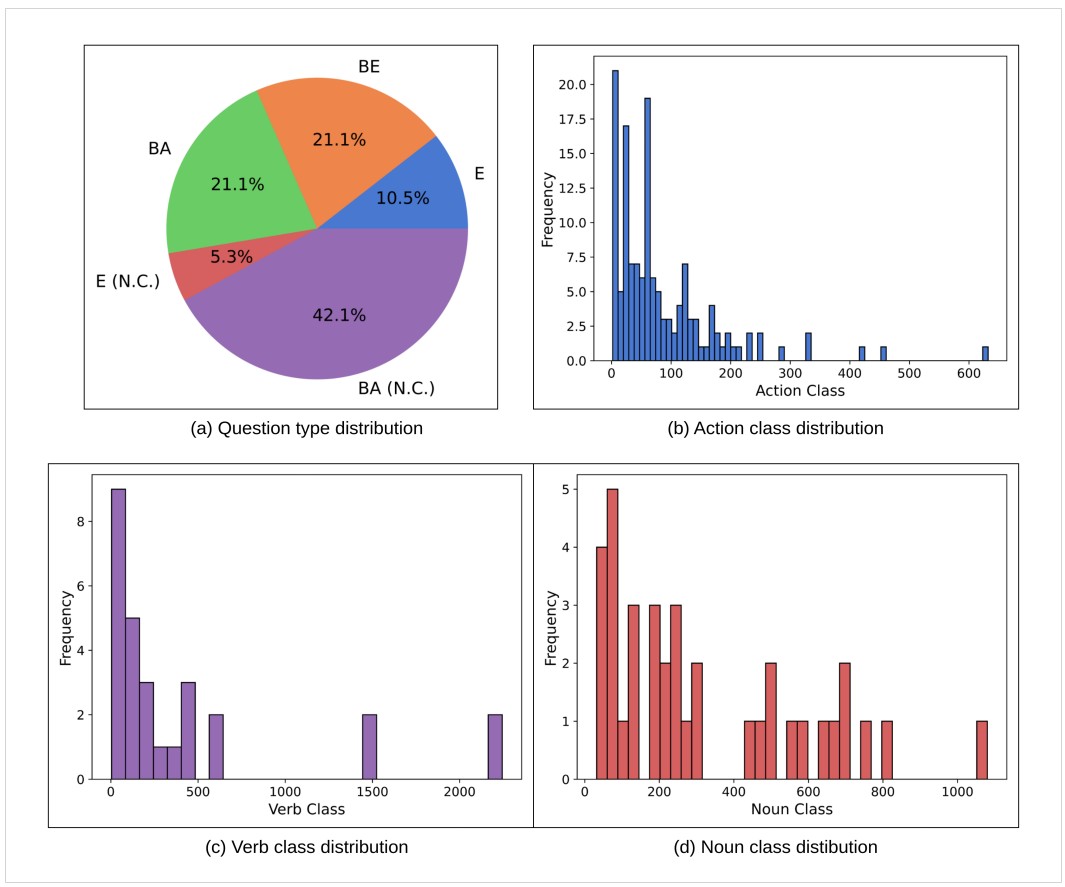

Figure 7: Metrics of the dataset (a) distribution of question types (same for training and testing set), (b) histogram plot of frequencies of action classes (c) histogram plot of frequencies of verb classes (d) histogram plot of frequencies of noun classes.

Table 14: Hyperparameters and checkpoint details of CLAVI finetuning experiment

| Model | Checkpoint | Epochs | LR |
|---|---|---|---|
| JustAsk | HowToVQA69M, WebVidVQA3M | 20 | 1.00E-05 |
| FrozenBiLM | WebVid10M | 20 | 5.00E-05 |
| All-In-One+ | Webvid, YT-Temporal-180M, HowTo100M, CC3M, CC12M, COCO, Visual Genome, SBU Captions | 10 | 1.00E-04 |
| Singularity-T | COCO, Visual Genome, SBU Captions, CC3M, CC12M, WebVid | 20 | 1.00E-05 |

Table 15: Fine-grained performance (% of accuracy) on CLAVI for question (Q) and counterfactual question (Q'), video (V) and counterfactual video (V') (Note: N.C. refers to Negative Control)

| V/V' | Question | Q/Q' | JustAsk | FrozenBiLM | Singularity-T | All-in-one+ |
|------|----------|------|---------|------------|---------------|-------------|
| V | E-type | Q | $89.55 \pm 0.01$ | $87.51 \pm 0.00$ | $90.75 \pm 0.03$ | $86.08 \pm 2.59$ |
| | E-type (N.C.) | - | $75.28 \pm 0.02$ | $88.66 \pm 0.00$ | $79.16 \pm 0.03$ | $69.34 \pm 11.72$ |
| | BE-type | Q | $69.80 \pm 0.07$ | $69.15 \pm 0.01$ | $98.23 \pm 0.01$ | $99.31 \pm 0.84$ |
| | | Q' | $30.58 \pm 0.07$ | $73.25 \pm 0.01$ | $1.87 \pm 0.01$ | $0.73 \pm 0.84$ |
| | BA-type | Q | $27.81 \pm 0.02$ | $56.88 \pm 0.01$ | $62.55 \pm 0.09$ | $25.82 \pm 5.49$ |
| | | Q' | $72.31 \pm 0.02$ | $86.79 \pm 0.01$ | $37.23 \pm 0.09$ | $74.31 \pm 0.84$ |
| | BA-type (N.C.) | - | $98.23 \pm 0.00$ | $96.79 \pm 0.00$ | $93.72 \pm 0.03$ | $98.44 \pm 1.02$ |
| V' | E-type | Q | $89.17 \pm 0.01$ | $86.96 \pm 0.01$ | $90.58 \pm 0.02$ | $86.03 \pm 2.66$ |
| | E-type (N.C.) | Q | $76.10 \pm 0.03$ | $88.45 \pm 0.01$ | $79.04 \pm 0.03$ | $69.17 \pm 11.26$ |
| | BE-type | Q | $30.18 \pm 0.07$ | $73.61 \pm 0.01$ | $1.80 \pm 0.01$ | $0.76 \pm 1.00$ |
| | | Q' | $69.88 \pm 0.07$ | $70.00 \pm 0.02$ | $98.28 \pm 0.01$ | $99.12 \pm 1.02$ |
| | BA-type | Q | $71.61 \pm 0.02$ | $85.43 \pm 0.01$ | $38.00 \pm 0.08$ | $74.24 \pm 5.12$ |
| | | Q' | $28.34 \pm 0.02$ | $54.44 \pm 0.00$ | $62.15 \pm 0.07$ | $25.90 \pm 4.93$ |
| | BA-type (N.C.) | - | $98.51 \pm 0.00$ | $96.87 \pm 0.00$ | $93.51 \pm 0.03$ | $98.46 \pm 1.04$ |

Table 16: Statistics of L2 distance values between aligned attention matrices of BA-type CLAVI questions, averaged over all heads and layers for FrozenBiLM. We report the statistics separately for correctly and incorrectly answered consistent counterfactual predictions.

| Type | Consistent Prediction | Mean | Variance |
|------|----------------------|------|----------|
| Video Counterfactual | Correct | 0.70 | 0.02 |
| | Incorrect | 0.50 | 0.03 |
| Text Counterfactual | Correct | 0.55 | 0.01 |
| | Incorrect | 0.38 | 0.02 |

### A.3.7 REPRESENTATION SENSITIVITY ANALYSIS

CLAVI can be used for diverse analyses to understand and interpret the joint multimodal representations in VideoQA models. We present one such analysis here. We want to find out the difference in representations between correctly and wrongly-answered counterfactual pairs. Ideally, the counterfactual pairs should have distinctly dissimilar representations to be answered correctly.

We use L2 norm as the distance metric. For CLAVI, we construct counterfactuals by augmenting the sequence of the frames (video counterfactuals) or replacing before/after and beginning/end (text counterfactual). Hence, we cannot directly compute the distance between the attention matrices of the counterfactuals because they contain different tokens (text counterfactual) or different order of the same tokens (video counterfactual). We solve this by finding token correspondence between the counterfactual pairs for each layer and head. By treating each attention matrix as a graph, we model the matrix alignment problem to finding the node correspondence between two isomorphic weighted directed complete graphs. Node correspondence between two graphs can be viewed as an instance of a linear sum assignment problem. That is, we want to learn a permutation transformation so that the two attention matrices as similar. We define similarity as negative of L2 distance. We solve this using modified Jonker-Volgenant algorithm as described by Crouse (2016).

We plot the histogram of L2 distance (averaged over heads and layers) for BA-type video and question counterfactuals in Figure 8. As expected, we find that if the answer is correct, then the average L2 distance is generally higher (skewed towards right). The mean and variance values L2 mean distribution of the correctly and incorrectly answered counterfactual pairs is summarized in Table 16. We find that the correct predictions have higher mean and lower variance than the incorrectly inferred counterfactual pairs. These findings validate that the model in indeed learning joint multimodal representations rather than creating its illusion.

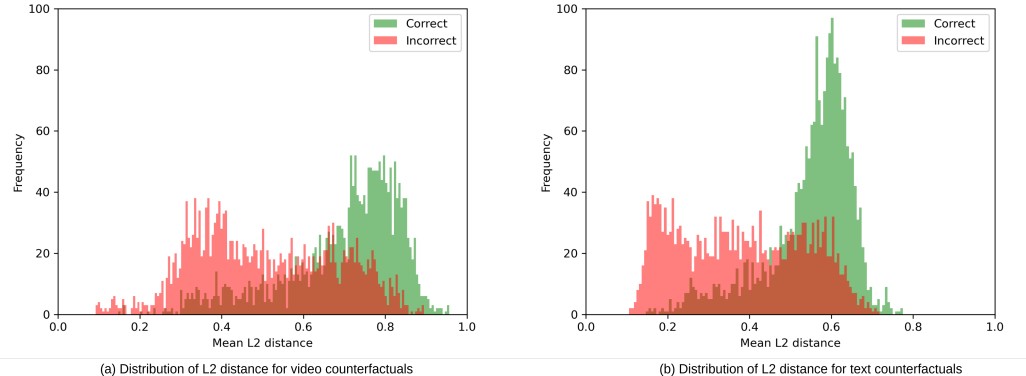

(a) Distribution of L2 distance for video counterfactuals    (b) Distribution of L2 distance for text counterfactuals

Figure 8: Histogram plots of mean l2 distance between counterfactual BA-type pairs for (a) video and (b) text counterfactuals for FrozenBiLM predictions (note that green is consistently correct; that is both the pairs in the counterfactuals are correctly answered, Similarly, red is consistently incorrect; that is at least one of the instance from the counterfactual pair is incorrectly answered).

### A.3.8   DATASHEET

In this section we provide a more detailed documentation of the dataset with the intended uses. We base ourselves on the datasheet proposed by Gebru et al. (2021)

Motivation

- **For what purpose was the dataset created?** CLAVI is curated to diagnose and benchmark the joint multimodal understanding in VideoQA models. It uses temporal counterfactuals in video and question domains to assess the impact of multimodal shortcuts that can create an illusion of joint temporal understanding.

- **Who created the dataset and on behalf of which entity?** ANONYMOUS

- **Who funded the creation of the dataset?** ANONYMOUS

Composition

- **What do the instances that comprise the dataset represent?** Each instance in CLAVI comprises of a video, question, question type, and answer ("yes" or "no").

- **How many instances are there in total?** CLAVI consists of 6,018 videos composing of 3,830 training and 2,188 testing videos. Each video is associated with 19 question-answer pairs, hence 114,342 data-points (72,770 training and 41,572 testing).

- **Does the dataset contain all possible instances or is it a sample (not necessarily random) of instances from a larger set?** The videos and question answer pairs in CLAVI are generated by manipulating real-world real-world videos. In theory, we can generate more instances with more videos with temporal annotations. We will release the code to generate the true and counterfactual video and question instances.

- **What data does each instance consist of?** Each instance in CLAVI comprises of a video, question, question type, and answer ("yes" or "no").

- **Is there a label or target associated with each instance?** Yes, each question is associated with a 'type' label depending on the type of the question (types described in the main manuscript).

- **Is any information missing from individual instances?** No, all the instances have complete information the corresponding attributes.

- **Are relationships between individual instances made explicit?** Yes, the *video_name* attribute if of the form XXXXXXXX_1 for the original video segment and XXXXXXXX_2

for the counterfactual video segment, where XXXXXXXX is a unique 8-digit video id. The relationship between counterfactual questions is tabulated in the README file of the dataset.

- **Are there recommended data splits (e.g., training, development/validation, testing)?** We provide the split files which are curated from the original split files of Charades.

- **Are there any errors, sources of noise, or redundancies in the dataset?** No. The owners of Charades do not report any known errors. And since our data is generated by machine, we do not expect any errors. For unforeseen errors in temporal annotation boundaries in the original dataset, we eliminate it by selecting the segments where the actions of interest do not occur in the immediate neighbourhood (detailed in the main manuscript).

- **Is the dataset self-contained, or does it link to or otherwise rely on external resources?** This dataset provides video IDs from the Charades dataset under their Non-Commercial license.

- **Does the dataset contain data that might be considered confidential?** No. We curate our dataset from publicly and non-commercially available Charades dataset.

- **Does the dataset identify any sub-populations (e.g., by age, gender)?** No. While it is possible to identify gender from Charades temporal captions, we do not use it in the curation of CLAVI. We only use neutral pronoun *someone*.

- **Is it possible to identify individuals (i.e., one or more natural persons), either directly or indirectly (i.e., in combination with other data) from the dataset?** No. The owners of Charades dataset have anonymized the subject information.

- **Does the dataset contain data that might be considered sensitive in any way?** No. The owners of Charades dataset ensure this and we curate CLAVI from Charades.

Collection Process

- **How was the data associated with each instance acquired?** Each sample of CLAVI associates with a question, answer (yes/no) and video-id from Charades dataset.
  We generated instances in CLAVI that with corresponding video, question, answer and their respective counterfactuals from Charades.

- **What mechanisms or procedures were used to collect the data (e.g., hardware apparatuses or sensors, manual human curation, software programs, software APIs)?** We design a template-based VideoQA generation process to generate data each instance from Charades.

- **Who was involved in the data collection process (e.g., students, crowdworkers, contractors) and how were they compensated (e.g., how much were crowdworkers paid)?** Not applicable

- **If the dataset is a sample from a larger set, what was the sampling strategy (e.g., deterministic, probabilistic with specific sampling probabilities)?** Yes. We have outlined the process of filtering the data in detail in the appendix.

- **Who was involved in the data collection process (e.g., students, crowdworkers, contractors) and how were they compensated (e.g., how much were crowdworkers paid)?** Not applicable.

- **Over what timeframe was the data collected?** Our dataset is generated from Charades. We generate the dataset from February 2023 to June 2023.

- **Were any ethical review processes conducted (e.g., by an institutional review board)?** Not Applicable.

- **Does the dataset relate to people?** Yes. The Charades dataset contains videos of humans performing actions and we use it to curate CLAVI under their non-commercial license. However, we do not use the information pertaining to humans anyway.

- **Did you collect the data from the individuals in question directly, or obtain it via third parties or other sources (e.g., websites)?** No. Our video data is from Charades under their Non-Commercial license. Charades Homepage: https://prior.allenai.org/projects/charades.

- **Were the individuals in question notified about the data collection?** Not applicable. We curate our dataset from Charades and the original owners have ensured this.

- **Did the individuals in question consent to the collection and use of their data?** Not applicable. We curate our dataset from Charades and the original owners have ensured this.

- **If consent was obtained, were the consenting individuals provided with a mechanism to revoke their consent in the future or for certain uses?** Not applicable.

Preprocessing/cleaning/labeling

- **Was any preprocessing/cleaning/labeling of the data done (e.g., discretization or bucketing, tokenization, part-of-speech tagging, SIFT feature extraction, removal of instances, processing of missing values)** No.

- **Was the "raw" data saved in addition to the preprocessed/cleaned/labeled data (e.g., to support unanticipated future uses)?** No. The raw data (Charades) is distributed under Non-Commerical license.

- **Is the software that was used to preprocess/clean/label the data available?** Not applicable.

Uses

- **Has the dataset been used for any tasks already?** Video Question Answering.

- **Is there a repository that links to any or all papers or systems that use the dataset?** No.

- **What (other) tasks could the dataset be used for?** Video-Text and Text-Video retrieval.

- **Is there anything about the composition of the dataset or the way it was collected and preprocessed/cleaned/labeled that might impact future uses?** No.

- **Are there tasks for which the dataset should not be used?** No.

- **Will the dataset be distributed to third parties outside of the entity (e.g., company, institution, organization) on behalf of which the dataset was created?** CLAVI is an academic dataset for public non-commercial use.

- **How will the dataset be distributed (e.g., tarball on website, API, GitHub)?** The dataset files will be released on GitHub.

- **When will the dataset be distributed?** Latest by the official paper acceptance.

- **Will the dataset be distributed under a copyright or other intellectual property (IP) license, and/or under applicable terms of use (ToU)?** Yes. The dataset will be released under GPL3.0 license and terms of usage will be outlined on the dataset hosting website along with the license and the required scripts.
  **Have any third parties imposed IP-based or other restrictions on the data associated with the instances?** No.
  **Do any export controls or other regulatory restrictions apply to the dataset or to individual instances?** No.

Maintenance

- **Who will be supporting/hosting/maintaining the dataset?** ANONYMOUS
  **How can the owner/curator/manager of the dataset be contacted (e.g., email address)?** ANONYMOUS

- **Is there an erratum?** Not yet.

- **Will the dataset be updated (e.g., to correct labeling errors, add new instances, delete instances)?** Updates, if any, will be clearly mentioned on GitHub.

- **If the dataset relates to people, are there applicable limits on the retention of the data associated with the instances (e.g., were the individuals in question told that their data would be retained for a fixed period of time and then deleted)?** No.

- **Will older versions of the dataset continue to be supported/hosted/maintained?** Yes.

- **If others want to extend/augment/build on/contribute to the dataset, is there a mechanism for them to do so?** Yes, we will provide the necessary code files with the dataset.

## A.4    LIMITATIONS AND FUTURE WORK

Our dataset is intentionally simple, so as to focus the benchmark only on simple temporal sequence understanding, which preempts spatio-temporal referential understanding. We plan to include more complex temporal organizations of action classes like containment and partial-overlap that are defined using prepositions like *during* and *while* in future work. As the current state-of-the-art models catch-up to our benchmark, our future plan is to curate a more complex dataset with more natural questions that include temporal referring expressions with similar balanced doubly-negative strategy.

