# OpenReview forum: "Revealing the Illusion of Joint Multimodal Understanding in VideoQA Models"
_ICLR.cc/2024/Conference — Submitted to ICLR 2024_

### Official Review · Reviewer_Vyie · 2023-10-28

**Soundness:** 3 good
**Presentation:** 3 good
**Contribution:** 3 good
**Rating:** 6
**Confidence:** 2

**Summary:**

This paper proposed a lightweight and non-parametric probe, to critically analyze multimodal representations. Then the paper proposed a diagnostic dataset for coupled multimodal understanding in VideoQA.

**Strengths:**

(1) The paper formulation is good and clear.

(2) The question that the paper tries to answer is meaningful.

**Weaknesses:**

(1) Did the authors conduct any ablation studies to isolate the influence stemming from the data itself rather than the methodology? For instance, exploring whether either video or text inherently poses greater learning challenges could provide valuable insights.

(2) Can these findings be extrapolated to other question-answer tasks, such as image-based question-answering?

**Questions:**

Please see the comments above.

---

> ### Author Response · Authors · 2023-11-20
> **Thanks and Reply to Reviewer Vyie**
>
> We thank the reviewer for their feedback. We have addressed their concerns below:
>
> > **W1** Text versus video modality {Did the...valuable insights?}
>
> **R1** Thanks for the question! We have analyzed the difference between attention scores learnt in the video-consistent and text-consistent pairs in CLAVI. Please refer to representation sensitivity analysis in A.3.7. Our results suggest that for FrozenBiLM, it learns more distinct representations for counterfactual videos than counterfactual text (Table 16, Figure 8). We leave studying this for more models and finetuning settings for our future work.
>
> > **W2** Extension to ImageQA
>
> **R2** Yes, QUAG is generic and can be directly applied to any other combination modalities. We leave it for the future work.

---

> > ### Author Response · Authors · 2023-11-22
> >
> > Dear Reviewer Vyie,
> >
> > We hope that our responses to your queries are satisfactory. A gentle reminder that we are keen to hear back your constructive reply. We would be happy to provide any further information and clarifications. We request you to consider increasing the assessment scores.
> >
> > Thanks,\
> > Authors

---

### Official Review · Reviewer_ABMn · 2023-10-30

**Soundness:** 2 fair
**Presentation:** 3 good
**Contribution:** 2 fair
**Rating:** 5
**Confidence:** 4

**Summary:**

The paper proposes to even out the attention weights responsible for individual modalities or for modality mixing in the attention modules of a multimodal transformer model, as a way to probe which (combination of) modalities contribute to making predictions. The paper uses 2 video QA models (JustAsk, FrozenBiLM) and 4 datasets in the study (ActivityNetQA, MSRVTT-QA, NextQA, ATP-Hard). The authors then propose a new dataset (CLAVI) containing binary video QAs with good balance across positive and negative answers to better showcase the shortcuts used by existing multimodal models; they fine-tune and evaluate 4 videoQA models (JustAsk, FrozenBilm, Singularity-T, All-in-One+) on the proposed dataset and discuss their weaknesses.

**Strengths:**

A better understanding of how multimodal models operate is of critical importance for the community. The idea of short-circuiting modalities is interesting.
Proposing more challenging and balanced benchmarks is very valuable to guide research.
The insight that multimodal models rely mostly on text to make predictions is valuable and shows that the video components of multimodal models need significant improvement.
The consistency metrics are a useful contribution.

**Weaknesses:**

A discussion about invasive vs non-invasing probing methods is needed. E.g. the authors should cite and discuss non-invasive analysis methods that rely on gradient backpropagation, e.g. MultiViz: Towards visualizing and understanding multimodal models, ICLR2023. I don’t know if our current understanding of deep models is good enough to perform invasive probing like the mechanism proposed here and draw strong conclusions from it. E.g. the authors replace all attention blocks in a model with the proposed modified blocks. But it is possible that not all blocks in the model behave in the same way, e.g. early-fusion vs late-fusion of modalities. Would it make sense to replace blocks progressively and see where the performance drops?

Some of the experiments are not very conclusive. E.g. in Table 1, the only clear result is that both models are significantly impaired in the video-only setting, but the short-circuiting results are not conclusive, especially for JustAsk where there is almost no difference across all SC setups.

I have strong concerns about the proposed benchmark. Permuting segments in a video creates temporal discontinuities that can be exploited by the models in unexpected ways, especially when fine-tuning the models on the benchmark. Why is fine-tuning needed at all? Zero-shot evaluation would be better, especially for the purposes of diagnosing a model.

Page 8, the authors say “to account for class imbalances in the answers” – are the positive vs negative QAs not balanced in the dataset?

Could there be ambiguities in the video-question pairs when the videos are altered? E.g. in the example shown with Fig 2, “holding on clothes” and “turning on a light”; when the altered video starts, the light is already on, so saying that the light was turned on at the beginning is not completely wrong.

Some questions might be ambiguous or not well defined, e.g. for the before-after negative control questions, the example in Table 2: since “washing mirror” never happens in the video, it could be ambiguous to ask about a before/after relation.

Could the authors justify the choice of the models? JustAsk, FrozenBiLM have been outperformed by several newer models, so it is not clear if the analysis still holds.

The naming “counterfactual” for questions whose answer is negative can be misleading. E.g. in CLEVRER or Perception Test, “counterfactual” is used for questions that require imagining a different sequence of events from a given state of the environment, “what would happen if…”

Missing references:
MultiViz: Towards visualizing and understanding multimodal models, ICLR2023
Perception Test: A diagnostic benchmark for multimodal video models, NeurIPS2023 Benchmarks

**Questions:**

See weaknesses above.

---

> ### Author Response · Authors · 2023-11-20
> **Thanks and Reply to Reviewer ABMn**
>
> We thank the reviewer for their time and feedback. We are glad that they found our work useful and interesting.
>
> > **W1.1** Discussion on probing methods like MultiViz {A discussion...ICLR 2023}
>
> **R1.1** Thanks for the feedback! We have included an addiitonal related works section in Appendix A.1 on Multimodal fusion interpretability and visualization.
>
> > **W1.2** Invasive probing and progressive-QUAG {I don't know...performance drop?}
>
> **R1.2** We believe that it is through probes like QUAG that can improve the understanding of deep models. In fact, other works, such as [1] also use similar formulation of bi-modal attention matrices by using gradient weighted attention maps for visualization. Since our results from QUAG are consistent with the results of QUAG-attention and CLAVI, we are confident of our assessment.
>
> Yes, it is quite possible that the blocks can behave differently. Note that progressive-QUAG won’t ablate the modality interactions completely because the downstream blocks would mix the tokens within the modality (through the residual input) but can provide insights about the relative importance of upstream fusion blocks. We provide the results for progressive-QUAG on FrozenBiLM on NeXT-QA and ActivityNet-QA datasets ([https://imgur.com/a/wmwI3KJ](https://imgur.com/a/wmwI3KJ), Appendix A.2.8). progressive-QUAG further underscores the importance of the first half of the fusion layer in the performance of the model. Overall, the results are consistent and supplement our findings.
>
> References:
> [1] Chefer, H. et al. "Generic attention-model explainability for interpreting bi-modal and encoder-decoder transformers."  ICCV 2021.
>
> > **W2** Inconclusive experiments {Some of...SC setups}
>
> **R2** The original dataset-centric belief was that shortcuts are spurious features present in a dataset. Rather, in our work, we present a unified dataset-model analysis of shortcuts as the spurious features present in a dataset that are learned by the model [2]. That is, just because the datasets have some biases, it is not necessary that the model learns and uses those to achieve high performance (Section 2.2) We summarize the short-circuiting operations that cause the drop in dataset-model combinations below (Uni is unimodal and cross is crossmodal SC):
>
> | |ActivityNetQA  |MSRVTT-QA  |NeXT-QA  |ATP-H |
> |-|-|-|-|-|
> |JustAsk   |\- |\-  |\-   |\-   |
> |FrozenBiLM|Text, Uni, **Cross**|Text, Uni, **Cross**|Text, Uni|Text, Uni|
>
> The idea is that if a specific token mixing operation is important, short-circuiting it should decrease the performance. As shown above, none of the models are learning and leveraging the core features from video modality interactions. JustAsk model does not exploit the rich multimodal structure of the datasets to answer those datasets. FrozenBiLM, however, consistently relies on unimodal and text modality interactions. However, for ActivityNet-QA and MSRVTT-QA (and not NeXT-QA), it also relies on crossmodal interactions. This means that FrozenBiLM does not learn rich crossmodal representations through NeXT-QA and ATP-Hard, but does learn it for ActivityNet-QA and MSRVTT-QA.
>
> References:
> [2] Murali, N. et al., "Shortcut Learning Through the Lens of Early Training Dynamics", Workshop on Spurious Correlations, Invariance, and Stability, ICML 2023
>
> > **W3.1** Exploiting temporal discontinuity {I have...the benchmark}
>
> **R3.1** Temporal discontinuity is not a problem for our setting because:
> 1. Most of the models operate on sparsely sampled frames (example,  3 frames for All-in-one model) but still manage to  achieve high performance on standard benchmarks. However, to further reduce the dependence on boundary, we only consider the instances in which the action classes are separated by at least 4 seconds and each action occurs for sufficiently long duration (average length: 7.7 seconds). Please refer to Appendix A.3.4 for detailed dataset metrics.
> 2. While curating the dataset, we ensure that each action class in a pair is composed of different objects and verbs (for example, *turning on light* and *holding some clothes*), so that swapping the order of events makes them loosely independent of each other. Please refer to Appendix A.3.1 for detailed dataset curation process.
> 3. We only consider the questions of existence, before/after, beginning/end types that do not necessarily require boundary information. Also, the balanced questions and videos ensure that the boundary information does not provide shortcut information.
>
> With sufficient (1) separation between the actions, (2) independence between the action classes, (3) balanced instances with simple questions, we ensure that the dataset is capable of diagnosing joint multimodal understanding in VideoQA models.
> (answer continued in the following comment)

---

> ### Author Response · Authors · 2023-11-20
>
> **R 3.1 (Cont)** CLAVI is a diagnostic dataset that penalizes shortcuts over joint multimodal understanding through consistent accuracy metrics.  This means that low consistent accuracy on CLAVI confirms the lack of joint multimodal understanding, which is the case for 3 out of the 4 models tested (i.e. Singularity-Temporal, All-in-one+, JustAsk).  The high CLAVI results are necessary but not sufficient to conclude that FrozenBiLM has joint multimodal understanding. Hence, by looking at QUAG results, we ensure that the relatively higher consistent performance of FrozenBiLM on CLAVI is indeed due to joint multimodal understanding (consistent accuracy of FrozenBiLM on QUAG drops significantly on multimodal impairment; Section 3.2, Paragraph 3).
>
> > **W 3.2** Why finetuning and not zeroshot? {why...a model}
>
> **R 3.2**  Zero-shot evaluation can be unfair for models that have been pretrained on datasets with imbalance instances of “yes” or “no”. This is well-documented in VQA [3]. We found similar problem with zero-shot evaluation:
>
> | |JustAsk|FrozenBiLM|Singularity-T|
> |-|-|-|-|
> |Balanced Acc  |49.7|50 |50 |
> |$CAcc_{V}$-counter|0.2 |0.1 |0|
> |$CAcc_{T}$-counter|0.8 |0.2|0 |
>
>
> In this case, JustAsk almost always selects “no”, while Singularity and FrozenBiLM almost always select “yes” (answer frequency bias). Hence, their balanced accuracies are nearly 50%.
>
> References:
> [3] Guo, Y. "Loss re-scaling VQA: Revisiting the language prior problem from a class-imbalance view." IEEE Transactions on Image Processing (2021).
>
> > **W4** Imbalance in CLAVI {to account...the dataset?}
>
> **R4** The refer balanced counterfactuals to refer that we exhaustively cover all the possible permutations of counterfactuals . That is, for a given temporal question Q and video V, and their counterfactuals Q’ and V’, CLAVI consists of all – (Q,V), (Q,V’), (Q’,V) and (Q’,V’) instances.
> The answers “yes” and “no” are not balanced because we also consider negative control questions, the answers to which are always “no” (Section 3.1, Paragraph 3, Page 7).
>
> > **W5** Ambiguity in CLAVI {could there...not wrong}
>
> **R5** CLAVI is curated such that all the information required to answer the question is present in the video. Our interpretation of *“Did turning on a light happen before holding on to clothes?”* is the event *“turning on a light”* happen before the event *“ holding on to clothes”* in the video. Since this interpretation is (1) consistent in the training and testing phases, and (2) we perform finetuning, the model should be able to understand the interpretation of the question unambiguously.
>
> > **W6** Hard-Negative Questions in CLAVI {some questions...relation}
>
> **R6** To ensure that there is no ambiguity in negative control, we randomly sample an action class that is not composed of any of the (i) verbs or (ii) objects that occur in the untrimmed video (original video). This scheme is consistently employed to curate the entire dataset. All the questions are designed ensuring that the information required to answer them is completely contained within the video. This is a consistent scheme in many popular datasets. For example, in Activitynet-QA, “is the boy in white squatting before diving?” the answer is “no” (question id: v_Bs3OMhhUlY4_3).
>
> > **W7** Model-choice justification {could the...still holds}
>
> **R7** We use JustAsk (ICCV'21 Oral) and FrozenBiLM (NeurIPS'22) because they are (i) amongst the standard baseline models in VideoQA, (ii) compute-friendly to experiment with, (iii) the code and checkpoints of many VideoQA models are not open-sourced (iv) JustAsk and FrozenBiLM still perform well on the datasets we considered and also on newer and challenging benchmarks [4]. We use these representative models to present the issue of the illusion of joint multimodal understanding in VideoQA models and verify the presence of illusion in newer models as well, using CLAVI.
>
> We have now also included the results for All-in-one+ (CVPR ‘23) on ActivityNet-QA and MSRVTT-QA datasets (Appendix A.2.7). It contain 1/8 parameters than that of FrozenBiLM. We could not finetune it on NeXT-QA because of paucity of time. Our findings are consistent with the authors' claims on improved multimodal understanding.  Our findings are consistent with the author’s claims on improved multimodal understanding. However, since the model just uses 3 randomly sampled frames, it might be insufficient to have high consistent accuracy on CLAVI.
>
> | | A-QA  | M-QA |
> | - | -| - |
> | Acc | 41.9 | 43.1|
> | vid_only |14.2 | 4.2|
> | text_only| 23.5 | 20.8|
> | SC: unimodal| 11.4 | 3.8|
> | SC: crossmodal | 20.6 | 27.6|
> | SC: video|19.2|12.7|
> | SC: text|5.6|7.3|
>
> References:
> [4] Xiao, J. et al, "Can I Trust Your Answer? Visually Grounded Video Question Answering." arXiv preprint arXiv:2309.01327 (2023).
>
> > **W8** Counterfactual definition {The naming...happen if}
>
> **R8** We have clarified our definition of counterfactual in the main text.
>
> We have also included the mentioned citations.

---

> > ### Author Response · Authors · 2023-11-22
> >
> > Dear Reviewer ABMn,
> >
> > We hope our responses were helpful in answering your constructive queries. Please let us know if you require more information or clarifications. If you're satisfied, we kindly ask for your consideration in re-evaluating the scores.
> >
> > Thanks,\
> > Authors

---

> > > ### Comment · Reviewer_ABMn · 2023-11-22
> > > **Reply to Authors' rebuttal**
> > >
> > > Thank you very much for the in depth reply and for running the additional experiments -- they strengthen the submission.
> > >
> > > Thank you for adding the 0-shot results. The fact that those models have been trained on datasets with unbalanced yes/no answers and the evaluation here has balanced yes/no answers does not make the evaluation unfair in my opinion. On the contrary, it is revealing of the issues in the original training sets of those models that should be fixed.
> > >
> > > However, I maintain my concerns about the dataset creation. Indeed, many current models sample frames randomly from videos, but I believe that to be a weakness (or a design choice) of those models, and not a justification for compromising on the temporal continuity of the videos. Given that CLAVI is presented as a video benchmark, we should be able to diagnose any type of video model on it (both frame-based models and models that actually leverage the temporal continuity). Unfortunately, I don't see any easy fix for this issue.
> > >
> > > I am inclined to keep my original score.

---

> ### Author Response · Authors · 2023-11-23
>
> Thank you for the response to our clarifications!
> We are glad that you found most of our responses satisfactory.
>
> **Zero-shot Experiment**\
> Yes, we agree that the large-scale training datasets are often biased, which causes these models to learn biased representations (which might ultimately percolate to the finetuned models, inadvertently promoting shortcut learning). Thanks for sharing your insights with us!
>
> **Clarifications on CLAVI**\
> We list some follow-up clarifications on CLAVI based on your response:
> - We believe that the **diagnostic datasets** are always designed with a specific use case. In our case, it is joint multimodal understanding. As we discussed in the paper,
>     - High performance on CLAVI is necessary but not sufficient condition for joint multimodal understanding in these models (and hence, we propose and verify that the performance of FrozenBiLM is indeed due to multimodal understanding using QUAG).
>     - Low performance on CLAVI is **sufficient** to conclude that the models are unable to possess joint multimodal understanding.
> - **Not all models operate on low sampling frame rates**: For example, for JustAsk model, the number of training frames is **640 frames** per video. However, even then the model is unable to achieve respectable consistent accuracy on CLAVI.
> -   **Many diagnostic datasets have "_unnatural_" elements**: For example, ActionBench [1] considers video reversal as one of the diagnostic task. Similarly, Cut-and-paste based datasets like [2,3], that technically introduce spatial discontinuities are popular for diagnosing biases. Since our diagnostic datasets is insightful and has similar formulation as other well-established diagnostic datasets, we request for a holistic reconsideration.
>
>
> We are happy that you found our other supporting experiments and answers satisfactory. We request for you sincere re-evaluation  of scores. Please let us know if you need any more clarifications.
>
> Reference: \
> [1] Paxion: Patching Action Knowledge in Video-Language Foundation Models (NeurIPS 2023, Spotlight)\
> [2] Putting visual object recognition in context (CVPR 2020)\
> [3] When Pigs Fly: Contextual Reasoning in Synthetic and Natural Scenes (ICCV 2021)

---

### Official Review · Reviewer_jfVh · 2023-11-01

**Soundness:** 2 fair
**Presentation:** 3 good
**Contribution:** 2 fair
**Rating:** 5
**Confidence:** 3

**Summary:**

The authors are motivated by the question of whether the good performance of VideoQA models is from the models themselves or whether the benchmarks are not thorough enough to measure the performance. To answer the question, the authors have proposed QUAG and QUAG-attention. Moreover, they curated a CLAVI dataset with temporal counterfactuals to measure the consistency in the VideoQA performances.

**Strengths:**

1. The authors have introduced interesting approaches to prove/diagnose VideoQA models.

2. In addition, they clearly showed their experimental details.

**Weaknesses:**

1. QUAG and QUAG-attention have been evaluated on two models only. The authors have analyzed the performance drops in these two models on four different datasets. However, it is insufficient to conclude that Short-circuit and QUAG-attention are efficient by testing two models only.

2. the authors use CLAVI to diagnose joint multimodal understanding. However, showing consistent performances on the dataset with *temporal counterfactuals* does not mean the model is free from shortcuts.

3. The manuscript is a bit difficult to follow. The authors have to polish the paper.

4. An alternative approach to diagnosing models is to evaluate the generalization ability (e.g. zero-shot settings). How the proposed probes are effective compared to the evaluation?

5. Minor concerns:
* page 2: Is the maximum input sequence lengths of the multimodal fusion module $l$?
* Citation formatting.

**Questions:**

1. What are the differences between `video-consistency` and `text-consistency`? Section 3.1 explains them as follows, but they look identical.
```
If the model predicts the answers for a given question correctly for both – the video and its counterfactual video, it is called video-consistent. Similarly, for a given video, if the model predicts the answers to the question and its counterfactual question correctly, it is called text-consistent.
```

**Details Of Ethics Concerns:**

No ethics review needed. The authors also discussed it in Section 6.

---

> ### Author Response · Authors · 2023-11-20
> **Thanks and Reply to Reviewer jfVh**
>
> We thank the reviewer for their feedback. We have addressed their concerns below:
>
> > **W1** QUAG and QUAG-attention have been evaluated on 2 models {QUAG...only}
>
> **R1** Our main claim is that QUAG and QUAG-attention are methods to effectively introduce the reliance of the model on its biased representations. The paper did not make any claims about efficiency.
>
> Shortcuts are (i) the spurious features in a dataset (ii) that are learnt by the model [1]. QUAG analyzes the dependence of a model on specific types of modality interaction (for example, cross-modal interaction) through ablations. Therefore, if a model manages to achieve high accuracy with QUAG, it is relying on shortcuts for its high accuracy. Hence, QUAG enables a combined dataset-model analysis. We chose 4 standard datasets (ActivityNet-QA, MSRVTT-QA, NeXT-QA, ATP-Hard) and 2 popular standard baseline models in VideoQA (JustAsk - ICCV’21 Oral; FrozenBiLM – NeurIPS’22). Therefore, we consider 2 models and 4 datasets, that is 8 instances, as a representative to expose that the performance of many dataset-model combinations do not rely on multimodal understanding.\
> Furthermore, we prove the mathematical correctness of QUAG (Section 2.3), and extend it to QUAG-attention that leverages the biased representations through skinny attention matrices, without significant performance drop; reasserting the correctness and efficacy of QUAG.\
> We have now also included the results for All-in-one+ (CVPR'23) on ActivityNet-QA and MSRVTT-QA datasets (Appendix A.2.7).  It contain 1/8 parameters than that of FrozenBiLM. We could not finetune it on NeXT-QA because of paucity of time. Our findings are consistent with the authors' claims on improved multimodal understanding. However, since the model just uses 3 randomly sampled frames, it might be insufficient to have high consistent accuracy on CLAVI.
>
> | | A-QA  | M-QA |
> | - | -| - |
> | Acc | 41.9 | 43.1|
> | vid_only |14.2 | 4.2|
> | text_only| 23.5 | 20.8|
> | SC: unimodal| 11.4 | 3.8|
> | SC: crossmodal | 20.6 | 27.6|
> | SC: video|19.2|12.7|
> | SC: text|5.6|7.3|
>
> References:
> [1] Murali, N. et al., "Shortcut Learning Through the Lens of Early Training Dynamics", Workshop on Spurious Correlations, Invariance, and Stability, ICML 2023
>
> > **W2** CLAVI and models are not free of shortcuts
>
> **R2** We do not claim that our CLAVI dataset is free of all the biases that can be leveraged as shortcuts. Rather, our claim is that CLAVI is a diagnostic dataset that penalizes shortcuts over joint multimodal understanding through consistent accuracy metrics. \
> This means that low consistent accuracy on CLAVI confirms the lack of joint multimodal understanding, which is the case for 3 out of the 4 models tested (i.e. Singularity-Temporal, All-in-one+, JustAsk). \
> The high CLAVI results are necessary but not sufficient to conclude that FrozenBiLM has joint multimodal understanding. Hence, by looking at QUAG results, we ensure that the relatively higher consistent performance of FrozenBiLM on CLAVI is indeed due to joint multimodal understanding (consistent accuracy of FrozenBiLM on QUAG drops significantly on multimodal impairment; Section 3.2, Paragraph 3).
>
> > **W3** Manuscript is difficult to follow
>
> **R3** Thanks for the feedback. Are you able to point out more specific issues, examples or paragraphs? We would be happy to polish further to improve the paper.
>
> > **W4** Difference between our approach and generalization methods
>
> **R4**  While the generalization abilities can be used for model diagnosis (say, through zero-shot evaluation), the additional testing data might also contain the biases [2] that the model learnt during pretraining (hence, they are shortcuts). Since the accuracy metrics don’t typically penalize the shortcut learning, these methods cannot be directly used for diagnosis. Consider the zero-shot results from CLAVI below. The balanced accuracy is close to random chance, 50%. This is because the models either always predict the answer as "yes" (JustAsk) or "no" (FrozenBiLM, Singularity-T) (ans frequency bias). The consistent accuracy on the counterfactual subset confidently unveils the illusion of joint multimodal understanding in the model.
>
> | |JustAsk|FrozenBiLM|Singularity-T|
> |-|-|-|-|
> |Balanced Acc  |49.7|50 |50 |
> |$CAcc_{V}$-counter|0.2 |0.1 |0|
> |$CAcc_{T}$-counter|0.8 |0.2|0 |
>
> Furthermore, QUAG and QUAG-Attention, unlike generalization methods (i) do not rely on any external dataset, and (ii) pinpoints the exact failure modes in the models.
>
> References: [2] Guo, Y. "Loss re-scaling VQA: Revisiting the language prior problem from a class-imbalance view." IEEE Transactions on Image Processing (2021).
>
> > **W5.1** Is $l$ the maximum input sequence lengths of the multimodal fusion module?
>
> **R5.1** Yes. We have now clarified it in the main text.
>
> > **W5.2** Citation formatting
>
> **R5.2** We changed it to improve the readability. We have changed it back.

---

> ### Author Response · Authors · 2023-11-20
>
> > **Q1** Difference between video and text consistent acc?
>
> **A1** We explain it mathematically here. Consider a video V and a question Q, with answer $A_{V,Q}$. Let V’ and Q’ be the counterfactual video and question respectively, and F be the VideoQA model that maps a video and question pair to an answer. Then, we define video consistency as: {(F(V, Q) == $A_{V,Q}$) AND (F(V’,Q) == $A_{V’,Q}$)}. Similarly, text-consistency is defined as: {(F(V, Q) == $A_{V,Q}$) AND (F(V,Q’) == $A_{V,Q’}$)}. We have clarified it in the main text as well.

---

> > ### Author Response · Authors · 2023-11-22
> >
> > Dear Reviewer jfVh,
> >
> > We hope our responses have answered your queries. Please let us know if you have any more queries; we would be happy to provide more information on our work. We kindly request for your consideration in increasing the assessment scores.
> >
> > Thanks,\
> > Authors

---

> > > ### Comment · Reviewer_jfVh · 2023-11-23
> > >
> > > Thanks for providing the responses to the additional experiments. I have raised my rating from 3 to 5 as some of my concerns have been addressed.

---

> > > > ### Author Response · Authors · 2023-11-23
> > > >
> > > > We sincerely thank the reviewer for the reassessment of the scores. Do you have any unaddressed concerns that we can clarify to improve your assessment of our work? We would be glad to provide clarifications.

---

### Official Review · Reviewer_Ui4x · 2023-11-01

**Soundness:** 2 fair
**Presentation:** 3 good
**Contribution:** 2 fair
**Rating:** 5
**Confidence:** 5

**Summary:**

The paper investigates the true impact of modalities, i.e., visual and text, on Transformer-based multimodal models for video question answer (VideoQA) tasks. The authors aim to show that many fusion models based on multiple modalities are faced with being suboptimal and the performance improvement in these models may not truly rely on the multimodal representations contrary to what they claimed. For assessing the model reliance for multimodal learning, the authors introduce a simple Quadrant Average (QUAG) operator, and a new dataset called CLAVI that is gathered as a subset of the Charades Dataset. Moreover, they emphasize the importance of new accuracy-related metrics they introduced for assessing joint multimodal learning. In the experimental evaluation, some existing models are assessed via QUAG in available benchmark datasets and additional experiments are conducted on CLAVI with baseline models.

**Strengths:**

-The paper introduces an averaging-based simple technique to assess the impact of uni-modal, cross-modal representation in multimodal learning.

-They mention the importance dataset for VideoQA and introduce a new data collection. The question types in the introduced subsets are interesting.

**Weaknesses:**

-The QUAG is mainly not novel as it is already computed in self-attention based fusion transformers. It is simply the averaging over submatrices and here is simply used to investigate the impact of uni/cross modality. Similar techniques, such as averaging, are already used to visualize the multimodal representations upon training. I think the technique is not novel in this aspect.

-The fusion transformers can be designed in various ways. This study focuses on self-attention based fusion blocks, but cross-attention can be another direction. Particularly, the motivation of authors in selecting self-attention based fusion transformers is not clear and should be supported.

-I think some claims such as "FrozenBiLM consistently does not rely much on the video modality" are not justified well. Many observations may still related to dataset bias rather than the model bias. Therefore, datasets should be investigated for VideoQA task. As gathering data collections hard, reporting on various datasets is a feasible way as the SOTA research conducted.

-The authors gather a new dataset collection as a subset of Charades. The questions on ordering are interesting. However, they create new video samples by simply changing the order of video segments. This looks confusing as the temporal ordering of frames and transition from one activity to another is important (boundary cue). If we think simply that each frame is represented as a token, the ordering of these frames during transition is also important to answer complex questions. I think the dataset design is not so strong in this aspect.

**Questions:**

-Regarding indices used for QUAG operator on page 3: k looks like iterating over j indices (as j is in {q1 .. q2}) but the final output R(Z,W) is shown with ij indices. Can you check the correctness? Moreover, the s_ii is in {TT,TV,VT,VV} but in the above equation, it is used in a range [s_1 ... s_m]. Are the TT, TV etc. explained?

-The averaging is used to fuse token representations. However, there are other ways that can be easily integrated into transformers, such as CLS tokens. Do the authors investigate the usage of CLS for the same purpose?

-What is the targeted training setting in this paper for all models? For instance, the frozenBiLM reports results for fine-tuned, few-shot, and zero-shot cases.

---

> ### Author Response · Authors · 2023-11-20
> **Thanks and Reply to Reviewer Ui4x**
>
> Thanks for the feedback on our work. We are glad that you found our work interesting. We have addressed the concerns below:
>
> > **W1** QUAG is not novel {The QUAG... is not novel in this aspect}
>
> **R1**: The novelty of QUAG is its interpretability insight. While previous works have used averaging for either token pruning (for example, TESTA [1]) or multimodal visualization studies [2] (for example, VL-Interpret), our (i) intention, (ii) method and insights are substantially different and insightful from the previous works:\
> **Intention:**
> Shortcuts are (i) the spurious features in a dataset (ii) that are learnt by the model [3]. QUAG analyzes the dependence of a model on specific types of modality interaction (for example, cross-modal interaction) through ablations. Therefore, if a model manages to achieve high accuracy with QUAG, it is relying on shortcuts for its high accuracy (Section 2.2). Since we impair token mixing without any finetuning, we do not augment the learnt weights (representation) of the model. This facilitates analyzing the representations learnt by a model on a dataset.\
> **Method and insights:**  Multimodal visualization studies like [2] averages individual attention sub-matrices to classify the head as uni/crossmodal to visualize feature importance.
> 1. Our first contribution in QUAG is that we show and prove that ***it is the consistent row-wise averaging of the sub-matrices across all the heads and layers*** that has an effect of impairment (that is, short-circuiting). Replacing an entire submatrix with an average value of the submatrix (and not, row-wise average), as in [2], cannot faithfully short-circuit specific modality, as explained in Section 2.3. While the visualization methods can provide some interesting qualitative results, to the best of our knowledge, QUAG is the only method that provides quantitative dependence on modality interactions through their ablation.
> 2. Our second contribution is QUAG-attention that calculates attention on already-averaged keys to reduce the size of the attention matrix with (i) hardly any drop in performance, and (ii) not relying on “already computed <attention scores> in self-attention based fusion transformers”. Hence, (i) without any finetuning and optimization for performance, and (2) leveraging  the biased representations learnt by the model using first-principle analysis of self-attention, we prune the size of the attention matrix by as less as 45% and as much as 90% (assuming $L_\mathcal{V} = L_\mathcal{T} = 10$) with minimal performance drop.
>
> References\
> [1] Ren, S. et al., "TESTA: Temporal-Spatial Token Aggregation for Long-form Video-Language Understanding", EMNLP 2023\
> [2] Aflalo, E et al., "VL-interpret: An interactive visualization tool for interpreting vision-language transformers", CVPR 2022.\
> [3] Murali, N. et al., "Shortcut Learning Through the Lens of Early Training Dynamics", Workshop on Spurious Correlations, Invariance, and Stability, ICML 2023
>
> > **W2** Applicability to cross-attention {The fusion transformers...be supported}
>
> **R2**  We chose self-attention for our analysis because self-attention (1) is present in almost all fusion architectures, (2) and both the modalities are interleaved in the attention operation. Hence, many popular interpretability methods like VL-InterpreT [2], attention rollout, attention flow [4], commonly employed in visualization-based interpretability are also based on self-attention architectures. In case of cross-attention, the modality interactions are neatly segregated (TV, VT) into distinct attention units, and hence it becomes a trivial case of our analysis.
>
> References:
> [4] Samira A. et al.,. Quantifying Attention Flow in Transformers. ACL 2020
>
> > **W3** FrozenBilM results {I think some claims...research conducted}
>
> **R3**  The rationale, as mentioned in our comment on intention (rebuttal R1), is that the shortcuts are a function of **both – the model and the dataset**. We wanted to communicate that the spurious features that are present in the dataset are being learnt by FrozenBiLM, causing it to achieve high performance without relying on the core features present in the video modality. Thanks for pointing it out. We have replaced the statement to make it clearer. In fact our motivation to introduce CLAVI was that  “many observations may still be related to dataset bias rather than the model bias” (Section 3, titled “Does multimodal sub-optimality stems from dataset biases?”). CLAVI warrants joint multimodal understanding and penalizes shortcut learning.

---

> ### Author Response · Authors · 2023-11-20
>
> > **W4** Boundary information in CLAVI {The authors gather...not so strong in this aspect.}
>
> **R4** Thank you for the feedback. Boundary cues do not create a problem for our setting because:
> 1. As you correctly mentioned, most of the models operate on sparsely sampled frames (example, 3 frames for All-in-one model) but still manage to achieve high performance on standard benchmarks. However, to further reduce the dependence on boundary, we only consider the instances in which the action classes are separated by at least 4 seconds and each action occurs for sufficiently long duration (average length: 7.7 seconds). Please refer to Appendix A.3.4 for detailed dataset metrics.
> 2. While curating the dataset, we ensure that each action class in a pair is composed of different objects and verbs (for example, *turning on light* and *holding some clothes*), so that swapping the order of events makes them loosely independent of each other. Please refer to Appendix A.3.1 for detailed dataset curation process.
> 3. We only consider the questions of existence, before/after, beginning/end types that do not necessarily require boundary information. Also, the balanced questions and videos ensure that the boundary information does not provide shortcut information.
>
> With sufficient (1) separation between the actions, (2) independence between the action classes, (3) balanced instances with simple questions, we ensure that the dataset is capable of diagnosing joint multimodal understanding in VideoQA models.
>
> **[ADDENDUM]** We list additional points on discontinuity in CLAVI based on other reviews:
>
> - We believe that the **diagnostic datasets are always designed with a specific use case**. In our case, it is joint multimodal understanding. As we discussed in the paper,
>     - High performance on CLAVI is **necessary but not sufficient** condition for joint multimodal understanding in these models (and hence, we propose and verify that the performance of FrozenBiLM is indeed due to multimodal understanding using QUAG).
>     - Low performance on CLAVI is **sufficient** to conclude that the models are unable to possess joint multimodal understanding.
> - **Not all models operate on low sampling frame rates**: For example, for JustAsk model, the number of training frames is 640 frames per video but 3 for All-in-one+. However, even then the models are unable to achieve respectable consistent accuracy on CLAVI.
> - **Many diagnostic datasets have "unnatural" elements**: For example, ActionBench [5] considers video reversal as one of the diagnostic task. Similarly, Cut-and-paste based datasets like [6,7], that technically introduce spatial discontinuities are popular for diagnosing biases.
>
> Reference:\
> [5] Paxion: Patching Action Knowledge in Video-Language Foundation Models (NeurIPS 2023, Spotlight)\
> [6] Putting visual object recognition in context (CVPR 2020)\
> [7] When Pigs Fly: Contextual Reasoning in Synthetic and Natural Scenes (ICCV 2021)
>
>
>
> > **Q1** Clarification in Section 2.1 {Regarding indices used for...TT, TV etc. explained?}
>
> **A1** Yes, we have verified that the formulation is correct. $[R(Z,W)]\_{ij}$  refers to the element $(i,j)$ in the row-wise transformed matrix, $R(Z,W)$ . Hence, to calculate $R(Z,W)$ , the definition needs to be applied $i \times j$ times.  If $(i,j)$ does not lie in the submatrix W,  the value of  $[R(Z,W)]\_{ij}$ is unchanged, that is $[Z]\_{ij}$.
> However, if $(i,j)$ lie in the sub-matrix, perform row-wise averaging for the submatrix. That is, for the given row $i$, we take the mean of all the values in that row, indexed by k. Since we are averaging only across the columns in the sub-matrix, we only iterate over columns. Note that this can be efficiently implemented as a batched-matrix operation in pytorch. \
> We apologize for the confusion. $[s_1 ... s_m]$ does not refer to a range but a list of quadrants. Also, $\mathcal{T}\mathcal{T}$, $\mathcal{T}\mathcal{V}$,... etc refers to the quadrants corresponding to $A_{\mathcal{T}\mathcal{T}}$, $A_{\mathcal{T}\mathcal{V}}$ etc. We have clarified these in the main text.
>
> > **Q2** CLS token for fusion {the averaging...same purpose}
>
> **A2** We do not aim to compare fusion techniques. Rather, our intention is to use averaging of submatrices to disrupt the specific modality interactions (QUAG). It can be used with any of the fusion methods built on top of self-attention, like CLS-based. In fact, JustAsk model uses CLS token to infer the final answer. Using QUAG-attention on JustAsk, we are able to show that even after reducing the size of the attention matrix by 90% (that is retaining only the average text key and average video key) and saving 68% mulops, there is hardly any drop in the accuracy.
>
> > **Q3** Targeted setting for QUAG, CLAVI {what is...cases}
>
> **A3** For CLAVI, as mentioned in Section 3.2, we finetune all the models. However, for QUAG and QUAG-attention, as mentioned in Section 2.5, we use frozen models with the official dataset checkpoints.

---

> > ### Author Response · Authors · 2023-11-22
> >
> > Dear Reviewer Ui4x,
> >
> > We hope that our responses to your queries are satisfactory. A gentle reminder that we are keen to hearing back from you. We would be happy to provide any more information and clarifications. If you find our responses satisfactory, we kindly request you to increase the assessment rating.
> >
> > Thanks, \
> > Authors

---

> > > ### Comment · Reviewer_Ui4x · 2023-11-23
> > >
> > > Thanks for your detailed rebuttal which addresses many points I raised. But, I still have one major concern regarding dataset design. I will evaluate your rebuttal again and your answers to other reviewer's comments.

---

### Author Response · Authors · 2023-11-22
**[General Response]**

We thank all the reviewers for taking the time and effort to provide constructive feedback on our work. We are encouraged that the reviewers found our research question on assessing multimodal understanding in VideoQA models of **critical importance** (`ABMn`), and **meaningful** (`Vyie`). We are they enthused that they found QUAG, our interpretable method for quantifying the effect of specific modality interactions **interesting** (`jfVh`, `ABMn`). Our insights revealed that models rely on different modality interactions in different datasets, directing emphasis on combined dataset-model analysis of shortcut learning. In many cases, the models do not rely on the core features of the video modality for their high performance. Hence, we proposed CLAVI, an automatically curated diagnostic dataset that penalizes shortcut learning on joint multimodal understanding through consistency metrics. The reviewers found it to be **very valuable and challenging** (`ABMn`), and **interesting** (`Ui4x`). We are motivated that the reviewers recognized our **contributions useful** (`ABMn`) in underscoring the need for improvement in multimodal models.

We have incorporated the feedbacks of the reviewers add supplemented the response to their questions with the following additional experiments:
- Evaluation of QUAG on newer model: All-in-one+ (CVPR'23) on ActivityNetQA and MSRVTT-QA datasets (`jfVh`, `ABMn`)
- Progressive QUAG operation for analyzing the relative importance of fusion blocks (`ABMn`)
- Zero-shot performance on CLAVI dataset (`jfVh`, `ABMn`)

We are glad that the additional experiments support (infact, strengthen) our claims. Further, we have also made minor clarity corrections in the manuscript (`jfVh`, `Ui4x`). We are hoping to hear back from the reviewers soon and would be happy to take more questions and queries!

Thanks,\
Authors

---

### Meta-Review · Area_Chair_iszL · 2023-12-05

**Metareview:**

The authors study an important question: To which extent are the current VideoQA models making use of both modalities when making predictions? It might as well be that they are effectively ignoring the visual component and providing an illusion of joint understanding.

To this end the authors provide a new benchmark (CLAVI) which contains automatically generated balanced temporal counterfactuals in both question and video domains to accurately test if the models can jointly understand temporal cues in the question. The authors also develop a probe, QUAG, to demonstrate that high performance on established VideoQA benchmarks is not necessarily representative of faithful coupled multimodal understanding.

The authors found the topic and questions important. However, multiple reviewers pointed out flaws in the experimental setup and whether the empirical results support the strong claims put forth by the authors. While the rebuttal and discussion phase resulted in an increased score which makes the submission borderline, the reviewers and the AC ultimately remain unconvinced in the proposed method and whether the proposed dataset can be effectively used as advertised.

**Justification For Why Not Higher Score:**

Borderline: good problem and motivation, execution not good enough.

**Justification For Why Not Lower Score:**

N/A

---

### Decision · Program_Chairs · 2024-01-16

Reject